# CONTINUAL NORMALIZATION: RETHINKING BATCH NORMALIZATION FOR ONLINE CONTINUAL LEARNING

**Quang Pham**[1]**, Chenghao Liu**[2]**, Steven C.H. Hoi** [1,2]
[1] Singapore Management University
`hqpham.2017@smu.edu.sg`
[2] Salesforce Research Asia
`{chenghao.liu, shoi}@salesforce.com`

## ABSTRACT

Existing continual learning methods use Batch Normalization (BN) to facilitate training and improve generalization across tasks. However, the non-i.i.d and non-stationary nature of continual learning data, especially in the online setting, amplify the discrepancy between training and testing in BN and hinder the performance of older tasks. In this work, we study the cross-task normalization effect of BN in online continual learning where BN normalizes the testing data using moments biased towards the current task, resulting in higher catastrophic forgetting. This limitation motivates us to propose a simple yet effective method that we call Continual Normalization (CN) to facilitate training similar to BN while mitigating its negative effect. Extensive experiments on different continual learning algorithms and online scenarios show that CN is a direct replacement for BN and can provide substantial performance improvements. Our implementation is available at `https://github.com/phquang/Continual-Normalization`.

## 1 INTRODUCTION

Continual learning (Ring, 1997) is a promising approach towards building learning systems with human-like capabilities. Unlike traditional learning paradigms, continual learning methods observe a stream of tasks and simultaneously perform well on all tasks with limited access to previous data. Therefore, they have to achieve a good trade-off between retaining old knowledge (French, 1999) and acquiring new skills, which is referred to as the *stability-plasticity dilemma* (Abraham & Robins, 2005). Continual learning has been a challenging research problem, especially for deep neural networks because of their ubiquity and promising results on many applications (LeCun et al., 2015; Parisi et al., 2019).

While most previous works focus on developing strategies to alleviate catastrophic forgetting and facilitating knowledge transfer (Parisi et al., 2019), scant attention has been paid to the backbone they used. In standard backbone networks such as ResNets (He et al., 2016), it is natural to use Batch Normalization (BN) (Ioffe & Szegedy, 2015), which has enabled the deep learning community to make substantial progress in many applications (Huang et al., 2020). Although recent efforts have shown promising results in training deep networks without BN in a single task learning (Brock et al., 2021), we argue that BN has a huge impact on continual learning. Particularly, when using an episodic memory, BN improves knowledge sharing across tasks by allowing data of previous tasks to contribute to the normalization of current samples and vice versa. Unfortunately, in this work, we have explored a negative effect of BN that hinders its performance on older tasks and thus increases catastrophic forgetting. Explicitly, unlike the standard classification problems, data in continual learning arrived sequentially, which are *not independent and identically distributed* (*non-i.i.d*). Moreover, especially in the online setting (Lopez-Paz & Ranzato, 2017), the data distribution changes over time, which is also highly non-stationary. Together, such properties make the BN's running statistics heavily biased towards the current task. Consequently, during inference, the model normalizes previous tasks' data using the moments of the current task, which we refer to as the "cross-task normalization effect".

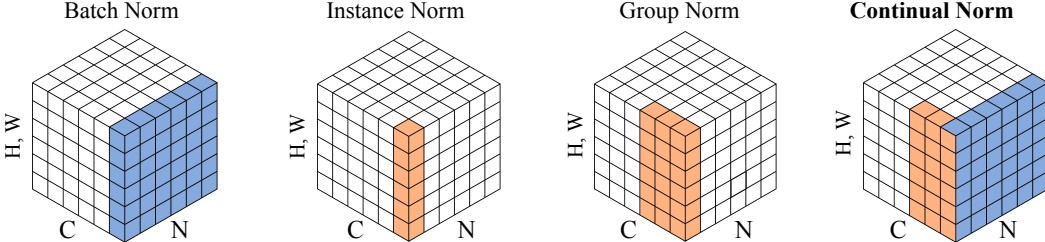

Figure 1: An illustration of different normalization methods using cube diagrams derived from Wu & He (2018). Each cube represents a feature map tensor with N as the batch axis, C as the channel axis, and (H,W) as the channel axes. Pixels in blue are normalized by the same moments calculated from different samples while pixels in orange are normalized by the same moments calculated from within sample. Best viewed in colors.

Although using an episodic memory can alleviate cross-task normalization in BN, it is not possible to fully mitigate this effect without storing all past data, which is against the continual learning purposes. On the other hand, spatial normalization layers such as GN (Wu & He, 2018) do not perform cross-task normalization because they normalize each feature individually along the spatial dimensions. As we will empirically verify in Section 5.1, although such layers suffer less forgetting than BN, they do not learn individual tasks as well as BN because they lack the knowledge transfer mechanism via normalizing along the mini-batch dimension. This result suggests that a continual learning normalization layer should balance normalizing along the mini-batch and spatial dimensions to achieve a good trade-off between knowledge transfer and alleviating forgetting. Such properties are crucial for continual learning, especially in the online setting (Lopez-Paz et al., 2017; Riemer et al., 2019), but not satisfied by existing normalization layers. Consequently, we propose Continual Normalization (CN), a novel normalization layer that takes into account the spatial information when normalizing along the mini-batch dimension. Therefore, our CN enjoys the benefits of BN while alleviating the cross-task normalization effect. Lastly, our study is orthogonal to the continual learning literature, and the proposed CN can be easily integrated into any existing methods to improve their performances across different online protocols.

In summary, we make the following contributions. First, we study the benefits of BN and its cross-task normalization effect in continual learning. Then, we identify the desiderata of a normalization layer for continual learning and propose CN, a novel normalization layer that improves the performance of existing continual learning methods. Lastly, we conduct extensive experiments to validate the benefits and drawbacks of BN, as well as the improvements of CN over BN.

## 2 NOTATIONS AND PRELIMINARIES

This section provides the necessary background of continual learning and normalization layers.

### 2.1 NOTATIONS

We focus on the image recognition problem and denote an image as $x \in \mathbb{R}^{W \times H \times C}$, where $W$, $H$, $C$ are the image width, height, and the number of channels respectively. A convolutional neural network (CNN) with parameter $\theta$ is denoted as $f_{\theta}(\cdot)$. A feature map of a mini-batch $B$ is defined as $a \in \mathbb{R}^{B \times C \times W \times H}$. Finally. each image $x$ is associated with a label $y \in \{1, 2 \dots, Y\}$ and a task identifier $t \in \{1, 2, \dots, T\}$, that is optionally revealed to the model, depending on the protocol.

### 2.2 CONTINUAL LEARNING

Continual learning aims at developing models that learn continuously over a stream of tasks. In literature, extensive efforts have been devoted to develop better continual learning algorithms, ranging from the dynamic architectures (Rusu et al., 2016; Serra et al., 2018; von Oswald et al., 2020; Yoon et al., 2018; Li et al., 2019; Xu & Zhu, 2018), experience replay (Lin, 1992; Chaudhry et al., 2019b; Aljundi et al., 2019a; Riemer et al., 2019; Rolnick et al., 2019; Wu et al., 2019), regularization (Kirkpatrick et al., 2017; Zenke et al., 2017; Aljundi et al., 2018; Ritter et al., 2018), to fast and slow frameworks (Pham et al., 2020; 2021). Our work takes an orthogonal approach by studying the benefits and drawbacks of BN, which is commonly used in most continual learning methods.

**Continual Learning Protocol** We focus on the *online continual learning* protocol (Lopez-Paz & Ranzato, 2017) over $T$ tasks. At each step, the model only observes training samples drawn from the current training task data and has to perform well on the testing data of all observed tasks so far. Moreover, data comes *strictly in a streaming manner*, resulting in a **single epoch** training setting. Throughout this work, we conduct experiments on different continual learning protocols, ranging from the task-incremental (Lopez-Paz & Ranzato, 2017), class-incremental (Rebuffi et al., 2017), to the task-free scenarios (Aljundi et al., 2019b).

## 2.3 NORMALIZATION LAYERS

Normalization layers are essential in training deep neural networks (Huang et al., 2020). Existing works have demonstrated that having a specific normalization layer tailored towards each learning problem is beneficial. However, the continual learning community stills mostly adopt the standard BN in their backbone network, and lack a systematic study regarding normalization layers (Lomonaco et al., 2020). To the best of our knowledge, this is the first work proposing a dedicated continual learning normalization layer.

In general, a normalization layer takes a mini-batch of feature maps $\boldsymbol{a} = (\boldsymbol{a}_1, \ldots, \boldsymbol{a}_B)$ as input and perform the Z-normalization as:

$$\boldsymbol{a}' = \boldsymbol{\gamma} \left( \frac{\boldsymbol{a} - \boldsymbol{\mu}}{\sqrt{\boldsymbol{\sigma}^2 + \epsilon}} \right) + \boldsymbol{\beta}, \tag{1}$$

where $\boldsymbol{\mu}$ and $\boldsymbol{\sigma}^2$ are the mean and variance calculated from the features in $B$, and $\epsilon$ is a small constant added to avoid division by zero. The affine transformation's parameters $\gamma, \beta \in \mathbb{R}^C$ are $|C|$-dimensional vectors learned by backpropagation to retain the layer's representation capacity. For brevity, we will use "moments" to refer to both the mean $\boldsymbol{\mu}$ and the variance $\boldsymbol{\sigma}^2$.

**Batch Normalization** BN is one of the first normalization layers that found success in a wide range of deep learning applications (Ioffe & Szegedy, 2015; Santurkar et al., 2018; Bjorck et al., 2018). During training, BN calculates the moments across the mini-batch dimension as:

$$\boldsymbol{\mu}_{BN} = \frac{1}{BHW} \sum_{b=1}^{B} \sum_{w=1}^{W} \sum_{H=1}^{H} \boldsymbol{a}_{bcwh}, \quad \boldsymbol{\sigma}_{BN}^2 = \frac{1}{BHW} \sum_{b=1}^{B} \sum_{w=1}^{W} \sum_{H=1}^{H} (\boldsymbol{a}_{bcwh} - \boldsymbol{\mu}_{BN})^2. \tag{2}$$

At test time, it is important for BN to be able to make predictions with only one data sample to make the prediction deterministic. As a result, BN replaces the mini-batch mean and variance in Eq. (2) by an estimate of the global values obtained during training as:

$$\boldsymbol{\mu} \leftarrow \boldsymbol{\mu} + \eta(\boldsymbol{\mu}_B - \boldsymbol{\mu}), \quad \boldsymbol{\sigma} \leftarrow \boldsymbol{\sigma} + \eta(\boldsymbol{\sigma}_B - \boldsymbol{\sigma}), \tag{3}$$

where $\eta$ is the exponential running average's momentum, which were set to 0.1.

**Spatial Normalization Layers** The discrepancy between training and testing in BN can be problematic when the training mini-batch size is small (Ioffe, 2017), or when the testing data distribution differs from the training distributions. In such scenarios, the running estimate of the mean and variance obtained during training are a poor estimate of the moments to normalize the testing data. Therefore, there have been tremendous efforts in developing alternatives to BN to address these challenges. One notable approach is normalizing along the spatial dimensions of each sample independently, which completely avoid the negative effect from having a biased global moments. In the following, we discuss popular examples of such layers.

**Group Normalization (GN) (Wu & He, 2018)** GN proposes to normalize each feature individually by dividing each channel into groups and then normalize each group separately. Moreover, GN does not normalize along the batch dimension, thus performs the same computation during training and testing. Computationally, GN first divides the channels $C$ into $G$ groups as:

$$\boldsymbol{a}'_{bgkhw} \leftarrow \boldsymbol{a}_{bchw}, \text{ where } k = \lfloor * \rfloor \frac{C}{G}, \tag{4}$$

Then, for each group $g$, the features are normalized with the moments calculated as:

$$\mu_{GN}^{(g)} = \frac{1}{m} \sum_{k=1}^{K} \sum_{w=1}^{W} \sum_{h=1}^{H} \boldsymbol{a}'_{bgkhw}, \quad \sigma_{GN}^{(g)^2} = \frac{1}{m} \sum_{k=1}^{K} \sum_{w=1}^{W} \sum_{h=1}^{H} (\boldsymbol{a}'_{bgkhw} - \mu_{GN}^{(g)})^2 \tag{5}$$

| Method | ACC | FM | LA | $\Delta_\mu^{(1)}$ | $\Delta_\mu^{(2)}$ | $\Delta_{\sigma^2}^{(1)}$ | $\Delta_{\sigma^2}^{(2)}$ |
|---|---|---|---|---|---|---|---|
| Single-BN | 72.81 | 18.65 | 87.74 | 10.85 | 0.91 | 3.69 | 7.74 |
| Single-BN* | **75.92** | **15.13** | **88.24** | | | | |
| ER-BN | 80.66 | 9.34 | 88.23 | 3.56 | 0.46 | 1.41 | 3.16 |
| ER-BN* | **81.75** | **8.51** | **88.46** | | | | |

Table 1: Evaluation metrics on the pMNIST benchmark. The magnitude of the differences are calculated as $\Delta_\omega^{(k)} = \|\omega_{BN}^{(k)} - \omega_{BN^*}^{(k)}\|_1$, where $\omega \in \{\mu, \sigma^2\}$ and $k \in \{1, 2\}$ denotes the first or second BN layer. Bold indicates the best scores.

GN has shown comparable performance to BN with large mini-batch sizes (e.g. 32 or more), while significantly outperformed BN with small mini-batch sizes (e.g. one or two). Notably, when putting all channels into a single group (setting $G = 1$), GN is equivalent to Layer Normalization (**LN**) (Ba et al., 2016), which normalizes the whole layer of each sample. On the other extreme, when separating each channel as a group (setting $G = C$), GN becomes Instance Normalization (**IN**) (Ulyanov et al., 2016), which normalizes the spatial dimension in each channel of each feature. Figure 1 provides an illustration of BN, GN, IN, and the proposed CN, which we will discuss in Section 4.

## 3 BATCH NORMALIZATION IN CONTINUAL LEARNING

### 3.1 THE BENEFITS OF NORMALIZATION LAYERS IN CONTINUAL LEARNING

We argue that BN is helpful for forward transfer in two aspects. First, BN makes the optimization landscape smoother (Santurkar et al., 2018), which allows the optimization of deep neural networks to converge faster and better (Bjorck et al., 2018). In continual learning, BN enables the model to learn individual tasks better than the no-normalization method. Second, with the episodic memory, BN uses data of the current task and previous tasks (in the memory) to update its running moments. Therefore, BN further facilitates forward knowledge transfer during experience replay: current task data is normalized using moments calculated from both current and previous samples. Compared to BN, spatial normalization layers (such as GN) lack the ability to facilitate forward transfer via normalizing using moments calculated from data of both old and current tasks. We will empirically verify the benefits of different normalization layers to continual learning in Section 5.1.

### 3.2 THE CROSS-TASK NORMALIZATION EFFECT

Recall that BN maintains a running estimate of the global moments to normalize the testing data. In the standard learning with a single task where data are i.i.d, one can expect such estimates can well-characterize the true, global moments, and can be used to normalize the testing data. However, online continual learning data is non-i.i.d and highly non stationary. Therefore, BN's estimation of the global moments is heavily biased towards the current task because the recent mini-batches only contain that task's data. As a result, during inference, when evaluating the older tasks, BN normalizes previous tasks' data using the current task's moments, which we refer to as the **cross-task normalization effect**.

We consider a toy experiment on the permuted MNIST (pMNIST) benchmark (Lopez-Paz & Ranzato, 2017) to explore the cross-task normalization effect in BN. We construct a sequence of five task, each has 2,000 training and 10,000 testing samples, and a multilayer perceptron(MLP) backbone configured as Input(784)-FC1(100)-BN1(100)-FC2(100)-BN(100)-Softmax(10), where the number inside the parentheses indicates the output dimension of that layer. We consider the Single and ER strategies for this experiment, where the Single strategy is the naive method that trains continuously without any memory or regularization. For each method, we implement an optimal-at-test-time BN variant (denoted by the suffix -BN*) that calculates the global moments using **all training data of all tasks** before testing. Compared to BN, BN* has the same parameters and only differs in the moments used to normalize the testing data. We emphasize that although BN* is unrealistic, it sheds light on how cross-task normalization affects the performance of CL algorithms. We report the averaged accuracy at the end of training ACC(↑) (Lopez-Paz et al., 2017), forgetting measure FM(↓) (Chaudhry et al., 2018), and learning accuracy LA(↑) (Riemer et al., 2019) in this experiment.

Table 1 reports the evaluation metrics of this experiment. Besides the standard metrics, we also report the moments' difference magnitudes between the standard BN and the global BN variant. We observe that the gap between two BN variants is significant without the episodic memory. When using the episodic memory, ER can reduce the gap because the pMNIST is quite simple and even the ER strategy can achieve close performances to the offline model (Chaudhry et al., 2019b). Moreover, it is important to note that training with BN on imbalanced data also affects the model's learning dynamics, resulting in a suboptimal parameter. Nevertheless, having an unbiased estimate of the global moments can greatly improve overall accuracy given the same model parameters. Moreover, this improvement is attributed to reducing forgetting rather than facilitating transfer: BN* has lower FM but almost similar LA compared to the traditional BN. In addition, we observe that the hidden features' variance becomes inaccurate compared to the global variance as we go to deeper layers ($\Delta_{\sigma^2}^{(1)} < \Delta_{\sigma^2}^{(2)}$). This result agrees with recent finding (Ramasesh et al., 2021) that deeper layers are more responsible for causing forgetting because their hidden representation deviates from the model trained on all tasks. Overall, these results show that normalizing older tasks using the current task's moments causes higher forgetting, which we refer to as the "cross-task normalization effect".

### 3.3 Desiderata for Continual Learning Normalization Layers

While BN facilitates continual learning, it also amplifies catastrophic forgetting by causing the cross-task normalization effect. To retain the BN's benefits while alleviating its drawbacks, we argue that an ideal normalization layer for continual learning should be adaptive by incorporating each feature's statistics into its normalization. Being adaptive can mitigate the cross-task normalization effect because each sample is now normalized differently at test time instead of being normalized by a set of biased moments. In literature, adaptive normalization layers have shown promising results when the mini-batch sizes are small (Wu & He, 2018), or when the number of training data is extremely limited (Bronskill et al., 2020). In such cases, normalizing along the spatial dimensions of each feature can alleviate the negative effect from an inaccurate estimate of the global moments. Inspired by this observation, we propose the desiderata for a continual learning normalization layer:

- Facilitates the performance of deep networks by improving knowledge sharing within and across-task (when the episodic memory is used), thus increasing the performance of all tasks (ACC in our work);
- Is adaptive at test time: each data sample should be normalized differently. Moreover, each data sample should contribute to its normalized feature's statistic, thus reducing catastrophic forgetting;
- Does not require additional input at test time such as the episodic memory, or the task identifier.

To simultaneously facilitate training and mitigating the cross-task normalization effect, a normalization layer has to balance between both across mini-batch normalization and within-sample normalization. As we will show in Section 5.1, BN can facilitate training by normalizing along the mini-batch dimension; however, it is not adaptive at test time and suffers from the cross-task normalization effect. On the other hand, GN is fully adaptive, but it does not facilitate training compared to BN. Therefore, it is imperative to balance both aspects to improve continual learning performance. Lastly, we expect a continual learning normalization layer can be a direct replacement for BN, thus, it should not require additional information to work, especially at test time.

## 4 Continual Normalization (CN)

CN works by first performing a spatial normalization on the feature map, which we choose to be group normalization. Then, the group-normalized features are further normalized by a batch normalization layer. Formally, given the input feature map $\boldsymbol{a}$, we denote $\text{BN}_{1,0}$ and $\text{GN}_{1,0}$ as the batch normalization and group normalization layers without the affine transformation parameters[1], CN obtains the normalized features $\boldsymbol{a}_{\text{CN}}$ as:

$$\boldsymbol{a}_{\text{GN}} \leftarrow \text{GN}_{1,0}(\boldsymbol{a}); \quad \boldsymbol{a}_{\text{CN}} \leftarrow \boldsymbol{\gamma}\text{BN}_{1,0}(\boldsymbol{a}_{\text{GN}}) + \boldsymbol{\beta}. \tag{6}$$

---

[1] which is equivalent to setting $\boldsymbol{\gamma} = 1$ and $\boldsymbol{\beta} = 0$

Table 2: Evaluation metrics of different normalization layers on the Split CIFAR100 and Split Mini IMN benchmarks. Bold indicates the best averaged scores, [†] suffix indicates non-adaptive method

| ER | Split CIFAR100 | | | Split Mini IMN | | |
|---|---|---|---|---|---|---|
| Norm. Layer | ACC($\uparrow$) | FM($\downarrow$) | LA($\uparrow$) | ACC($\uparrow$) | FM($\downarrow$) | LA($\uparrow$) |
| NoNL | 55.87±0.46 | 4.46±0.48 | 57.26±0.64 | 47.40±2.80 | 3.17±0.99 | 45.31±2.18 |
| BN[†] | 64.97±1.09 | 9.24±1.98 | 71.56±0.75 | 59.09±1.74 | 8.57±1.52 | 65.24±0.52 |
| BRN[†] | 63.47±1.33 | 8.43±1.03 | 69.83±2.52 | 54.55±2.70 | 6.66±1.84 | 58.53±1.88 |
| IN | 59.17±0.96 | 11.47±0.92 | 69.40±0.93 | 48.74±1.98 | 15.28±1.88 | 62.88±1.13 |
| GN | 63.42±0.92 | 7.39±1.24 | 68.03±0.19 | 55.65±2.92 | 8.31±1.00 | 59.25±0.72 |
| SN | 64.79±0.88 | 7.92±0.64 | 71.10±0.51 | 56.84±1.37 | 10.11±1.46 | 64.09±1.53 |
| CN(ours) | **67.48±0.81** | **7.29±1.59** | **74.27±0.36** | **64.28±1.49** | **8.08±1.18** | **70.90±1.16** |

| DER++ | Split CIFAR100 | | | Split Mini IMN | | |
|---|---|---|---|---|---|---|
| Norm. Layer | ACC($\uparrow$) | FM($\downarrow$) | LA($\uparrow$) | ACC($\uparrow$) | FM($\downarrow$) | LA($\uparrow$) |
| NoNL | 57.14±0.46 | 4.46±0.48 | 57.26±0.64 | 47.18±3.20 | 2.77±1.68 | 45.01±3.35 |
| BN[†] | 66.50±2.52 | 8.58±2.28 | 73.78±1.02 | 61.08±0.91 | 6.90±0.99 | 66.10±0.89 |
| BRN[†] | 66.89±1.22 | 6.98±2.23 | 73.30±0.08 | 57.37±1.75 | 6.66±1.84 | 66.53±1.56 |
| IN | 61.18±0.96 | 10.59±0.77 | 71.00±0.57 | 54.05±1.26 | 11.82±1.32 | 65.03±1.69 |
| GN | 66.58±0.27 | **5.70±0.69** | 69.63±1.12 | 60.50±1.91 | **6.17±1.28** | 63.10±1.53 |
| SN | 67.17±0.23 | 6.01±0.15 | 72.13±0.23 | 57.73±1.97 | 8.92±1.84 | 63.87±0.64 |
| CN(ours) | **69.13±0.56** | 6.48±0.81 | **74.89±0.40** | **66.29±1.11** | 6.47±1.46 | **71.75±0.68** |

In the first step, our GN component does not use the affine transformation to make the intermediate feature $BN_{1,0}(a_{GN})$ well-normalized across the mini-batch and spatial dimensions. Moreover, performing GN first allows the spatial-normalized features to contribute to the BN's running statistic, which further reduce the cross-task normalization effect. An illustration of CN is given in Figure 1.

By its design, one can see that CN satisfies the desiderata of a continual learning normalization layer. Particularly, CN balances between facilitating training and alleviating the cross-task normalizing effect by normalizing the input feature across mini-batch and individually. Therefore, CN is adaptive at test time and produces well-normalized features in the mini-batch and spatial dimensions, which strikes a great balance between BN and other instance-based normalizers. Lastly, CN uses the same input as conventional normalization layers and does not introduce extra learnable parameters that can be prone to catastrophic forgetting.

We now discuss why CN is a more suitable normalization layer for continual learning than recent advanced normalization layers. SwitchNorm (Luo et al., 2018) proposed to combine the moments from three normalization layers: BN, IN, and LN to normalize the feature map (Eq. 13). However, the blending weights $w$ and $w'$ make the output feature not well-normalized in any dimensions since such weights are smaller than one, which scales down the moments. Moreover, the choice of IN and LN may not be helpful for image recognition problems. Similar to SN, TaskNorm (Bronskill et al., 2020) combines the moments from BN and IN by a blending factor, which is learned for each task. As a result, TaskNorm also suffers in that its outputs are not well-normalized. Moreover, TaskNorm addresses the meta learning problem requires knowing the task identifier at test time, which violates our third criterion of requiring additional information compared to BN and our CN.

## 5 EXPERIMENT

We evaluate the proposed CN's performance compared to a suite of normalization layers with a focus on the online continual learning settings where it is more challenging to obtain a good global moments for BN. Our goal is to evaluate the following hypotheses: (i) BN can facilitate knowledge transfer better than spatial normalization layers such as GN and SN (ii) Spatial normalization layers have lower forgetting than BN; (iii) CN can improve over other normalization layers by reducing catastrophic forgetting and facilitating knowledge transfer; (iv) CN is a direct replacement of BN without additional parameters and minimal computational overhead. Due to space constraints, we briefly mention the setting before each experiment and provide the full details in Appendix D.

## 5.1 ONLINE TASK-INCREMENTAL CONTINUAL LEARNING

**Setting**    We first consider the standard online, task-incremental continual learning setting (Lopez-Paz & Ranzato, 2017) on the Split CIFAR-100 and Split Mini IMN benchmarks. We follow the standard setting in Chaudhry et al. (2019a) to split the original CIFAR100 (Krizhevsky & Hinton, 2009) or Mini IMN (Vinyals et al., 2016) datasets into a sequence of 20 tasks, three of which are used for hyper-parameter cross-validation, and the remaining 17 tasks are used for continual learning.

We consider two continual learning strategies: (i) Experience Replay (ER) (Chaudhry et al., 2019b); and (ii) Dark Experience Replay++ (DER++) (Buzzega et al., 2020). Besides the vanilla ER, we also consider DER++, a recent improved ER variants, to demonstrate that our proposed CN can work well across different ER-based strategies. All methods use a standard ResNet 18 backbone (He et al., 2016) (not pre-trained) and are optimized over one epoch with batch size 10 using the SGD optimizer. For each continual learning strategies, we compare our proposed CN with five competing normalization layers: (i) BN (Ioffe & Szegedy, 2015); (ii) Batch Renormalization (BRN) (Ioffe, 2017); (iii) IN (Ulyanov et al., 2016); (iv) GN (Wu & He, 2018); and (v) SN (Luo et al., 2018). We cross-validate and set the number of groups to be $G = 32$ for our CN and GN in this experiment.

**Results**    Table 2 reports the evaluation metrics of different normalization layers on the Split CIFAR-100 and Split Mini IMN benchmarks. We consider the averaged accuracy at the end of training ACC($\uparrow$) (Lopez-Paz et al., 2017), forgetting measure FM($\downarrow$) (Chaudhry et al., 2018), and learning accuracy LA($\uparrow$) (Riemer et al., 2019), details are provided in Appendix A. Clearly, IN does not perform well because it is not designed for image recognition problems. Compared to adaptive normalization methods such as GN and SN, BN suffers from more catastrophic forgetting (higher FM($\downarrow$) ) but at the same time can transfer knowledge better across tasks (higher LA($\uparrow$) ). Moreover, BRN performs worse than BN since it does not address the biased estimate of the global moments, which makes normalizing with the global moments during training ineffective. Overall, the results show that although traditional adaptive methods such as GN and SN do not suffer from the cross-task normalization effect and enjoy lower FM($\downarrow$) values, they lack the ability to facilitate knowledge transfer across tasks, which results in lower LA($\uparrow$) . Moreover, BN can facilitate knowledge sharing across tasks, but it suffers more from forgetting because of the cross-task normalization effect. Across all benchmarks, our CN comes out as a clear winner by achieving the best overall performance (ACC($\uparrow$) ). This result shows that CN can strike a great balance between reducing catastrophic forgetting and facilitating knowledge transfer to improve continual learning.

**Complexity Comparison**    We study the complexity of different normalization layers and reporting the training time on the Split CIFAR100 benchmark in Tabe 3. Both GN and our CN have minimal computational overhead compared to BN, while SN suffers from additional computational cost from calculating and normalizing with different sets of moments.

Table 3: Running time of ER on the Split CIFAR100 benchmarks of different normalization layers. $\Delta\%$ indicates the percentage increases of training time over BN

|  | BN | GN | SN | CN |
|---|---|---|---|---|
| Time (s) | 1583 | 1607 | 2036 | 1642 |
| $\Delta\%$ | 0% | 1.51% | 28.61% | 3.72% |

## 5.2 ONLINE CLASS-INCREMENTAL CONTINUAL LEARNING

**Setting**    We consider the online task incremental (Task-IL) and class incremental (Class-IL) learning problems on the Split-CIFAR10 (Split CIFAR-10) and Split-Tiny-ImageNet (Split Tiny IMN) benchmarks. We follow the same experiment setups as Buzzega et al. (2020) except the number of training epochs, which we set to *one*. All experiments uses the DER++ strategy (Buzzega et al., 2020) on a ResNet 18 (He et al., 2016) backbone (not pre-trained) trained with data augmentation using the SGD optimizer. We consider three different total episodic memory sizes of 500, 2560, and 5120, and focus on comparing BN with our proposed CN in this experiment.

**Result**    Table 4 reports the ACC and FM metrics on the Split CIFAR-10 and Split Tiny IMN benchmarks under both the Task-IL and Class-IL settings. For CN, we consider two configurations with the number of groups being $G = 8$ and $G = 32$. In most cases, our CN outperforms the standard BN on both metrics, with a more significant gap on larger memory sizes of 2560 and 5120. Interestingly, BN is highly unstable in the Split CIFAR-10, Class-IL benchmark with high standard deviations. In

Table 4: Evaluation metrics of DER++ with different normalization layers on the Split CIFAR-10 and Split Tiny IMN benchmarks. Parentheses indicates the number of groups in CN. Bold indicates the best averaged scores

| Buffer | Method | Split CIFAR-10 | | | | Split Tiny IMN | | | |
|---|---|---|---|---|---|---|---|---|---|
| | | Class-IL | | Task-IL | | Class-IL | | Task-IL | |
| | | ACC(↑) | FM(↓) | ACC(↑) | FM(↓) | ACC(↑) | FM(↓) | ACC(↑) | FM(↓) |
| 500 | BN | 48.9±4.5 | 33.6±5.8 | 82.6±2.3 | 3.2±1.9 | 6.7±0.1 | 38.1±0.5 | 40.2±0.9 | 6.5±1.0 |
| | CN ($G = 8$) | 48.9±0.3 | 27.9±5.0 | 84.7±0.5 | 2.2±0.3 | **7.3±0.9** | **37.5±2.3** | **42.2±2.1** | **4.9±2.2** |
| | CN ($G = 32$) | **51.7±1.9** | **28.2±4.0** | **86.2±2.2** | **2.0±1.3** | 6.5±0.7 | 40.1±1.7 | 40.6±1.4 | 6.8±1.9 |
| 2560 | BN | 52.3±4.6 | 29.7±6.1 | 86.6±1.6 | **0.9±0.7** | 11.2±2.3 | **36.0±2.0** | 50.8±1.7 | 2.8±1.2 |
| | CN ($G = 8$) | 53.7±3.4 | 25.5±4.7 | 87.3±2.7 | 1.6±1.6 | 10.7±1.3 | 38.0±1.2 | **52.6±1.2** | **1.5±0.5** |
| | CN ($G = 32$) | **57.3±2.0** | **21.6±5.6** | **88.4±1.1** | 1.7±1.1 | **11.9±0.3** | 36.7±1.2 | 51.5±0.2 | 2.8±0.9 |
| 5120 | BN | 52.0±7.8 | 26.7±10.3 | 85.6±3.3 | 2.0±1.5 | 11.2±2.7 | 36.8±2.0 | 52.2±1.7 | 2.6±1.5 |
| | CN ($G = 8$) | 54.1±4.0 | 24.0±4.0 | 87.1±2.8 | **0.7±0.7** | 12.2±0.6 | **34.6±2.6** | 53.1±1.8 | 3.1±1.9 |
| | CN ($G = 32$) | **57.9±4.1** | **22.2±1.0** | **88.3±0.9** | 1.3±0.9 | **12.2±0.2** | 35.6±1.3 | **54.9±1.5** | **1.5±1.1** |

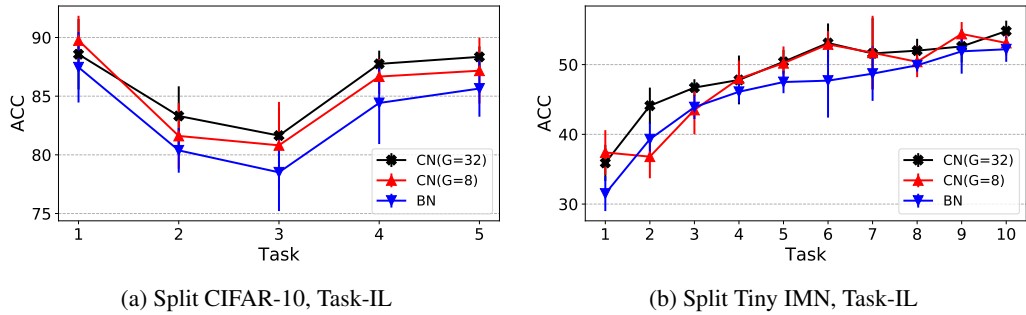

(a) Split CIFAR-10, Task-IL      (b) Split Tiny IMN, Task-IL

Figure 2: The evolution of ACC(↑) on observed tasks so far on the Split CIFAR-10 and Split Tiny IMN benchmarks, Task-IL screnario with DER++ and memory size of 5120 samples.

contrast, both CN variants show better and more stable results, especially in the more challenging Class-IL setting with a small memory size (indicated by small standard deviation values). We also report the evolution of ACC in Figure 2. Both CN variants consistently outperform BN throughout training, with only one exception at the second task on the Split Tiny IMN. CN ($G = 32$) shows more stable and better performances than BN and CN($G = 8$).

## 5.3 LONG-TAILED ONLINE CONTINUAL LEARNING

We now evaluate the normalization layers on the challenging task-free, long-tailed continual learning setting (Kim et al., 2020), which is more challenging and realistic since real-world data usually follow long-tailed distributions. We consider the PRS strategy (Kim et al., 2020) and the COCOseq and NUS-WIDEseq benchmarks, which consists of four and six tasks, respectively. Unlike the previous benchmarks, images in the COCOseq and NUS-WIDEseq benchmarks can have multiple labels, resulting in a long-tailed distribution over each task's image label. Following Kim et al. (2020), we report the average overall F1 (O-F1), per-class F1 (C-F1), and the mean average precision (mAP) at the end of training and their corresponding forgetting measures (FM). We also report each metric over the minority classes (<200 samples), moderate classes (200-900 samples), majority classes (>900 samples), and all classes. Empirically, we found that smaller groups helped in the long-tailed setting because the moments were calculated over more channels, reducing the dominants of head classes. Therefore, we use $G = 4$ groups in this experiment.

We replicate PRS with BN to compare with our CN and report the results in Table 5 and Table 6. We observe consistent improvements over BN from only changing the normalization layers, especially in reducing FM(↓) across all classes.

Table 5: Evaluation metrics of the PRS strategy on the COCOseq and NUS-WIDEseq benchmarks. We report the mean performance over five runs. Bold indicates the best averaged scores

| COCOseq | Majority | | | Moderate | | | Minority | | | Overall | | |
|---|---|---|---|---|---|---|---|---|---|---|---|---|
| | C-F1 | O-F1 | mAP | C-F1 | O-F1 | mAP | C-F1 | O-F1 | mAP | C-F1 | O-F1 | mAP |
| BN | 64.2 | 58.3 | 66.2 | 51.2 | 48.1 | **55.7** | 31.6 | 31.4 | 38.1 | 51.8 | 48.6 | 54.9 |
| CN(ours) | **64.8** | **58.5** | **66.8** | **52.5** | **49.2** | 55.7 | **35.7** | **35.5** | **38.4** | **53.1** | **49.8** | **55.1** |

| NUS-WIDEseq | Majority | | | Moderate | | | Minority | | | Overall | | |
|---|---|---|---|---|---|---|---|---|---|---|---|---|
| | C-F1 | O-F1 | mAP | C-F1 | O-F1 | mAP | C-F1 | O-F1 | mAP | C-F1 | O-F1 | mAP |
| BN | 24.3 | 16.1 | 21.2 | 16.2 | 16.5 | 20.9 | **28.5** | **28.2** | **32.3** | 23.4 | 20.9 | 25.7 |
| CN(ours) | **25.0** | **17.2** | **22.7** | **17.1** | **17.4** | **21.5** | 27.3 | 27.0 | 31.0 | **23.5** | **21.3** | **25.9** |

Table 6: Forgetting measure (FM($\downarrow$) ) of each metric from the PRS strategy on the COCOseq and NUS-WIDEseq benchmarks, lower is better. We report the mean performance over five runs. Bold indicates the best averaged scores

| COCOseq | Majority | | | Moderate | | | Minority | | | Overall | | |
|---|---|---|---|---|---|---|---|---|---|---|---|---|
| | C-F1 | O-F1 | mAP | C-F1 | O-F1 | mAP | C-F1 | O-F1 | mAP | C-F1 | O-F1 | mAP |
| BN | **23.5** | **22.8** | 8.4 | 30.0 | 30.2 | 9.4 | 36.2 | 35.7 | 13.2 | 29.7 | 29.5 | 9.7 |
| CN(ours) | **23.5** | 23.1 | **6.5** | **26.9** | **26.9** | **7.4** | **26.8** | **26.5** | **12.2** | **25.6** | **25.7** | **8.0** |

| NUS-WIDEseq | Majority | | | Moderate | | | Minority | | | Overall | | |
|---|---|---|---|---|---|---|---|---|---|---|---|---|
| | C-F1 | O-F1 | mAP | C-F1 | O-F1 | mAP | C-F1 | O-F1 | mAP | C-F1 | O-F1 | mAP |
| BN | 54.6 | 50.7 | 12.5 | 62.2 | 61.9 | 15.5 | 52.5 | 52.4 | 12.2 | 57.6 | 55.7 | 11.2 |
| CN(ours) | **52.7** | **48.7** | **8.6** | **58.8** | **58.0** | **10.3** | **51.2** | **50.6** | **11.6** | **57.4** | **55.5** | **8.1** |

## 5.4 Discussion of The Results

Our experiments have shown promising results for CN being a potential replacement for BN in online continual learning. While the results are generally consistent, there are a few scenarios where CN does not perform as good as other baselines. First, from the task-incremental experiment in Table 2, DER++ with CN achieved lower FM compared to GN. The reason could be from the DER++'s soft-label loss, which together with GN, overemphasizes on reducing FM and achieved lower FM. On the other hand, CN has to balance between reducing FM and improving LA. Second, training with data augmentation in the online setting could induce high variations across different runs. Table 4 shows that most methods have high standard deviations on the Split CIFAR-10 benchmark, especially with small memory sizes. In such scenarios, there could be insignificant differences between the first and second best methods. Also, on the NUS-WIDEseq benchmark, CN has lower evaluation metrics on minority classes than BN. One possible reason is the noisy nature of the NUS-WIDE dataset, including the background diversity and huge number of labels per image, which could highly impact the tail classes' performance. Lastly, CN introduces an additional hyper-parameter (number of groups), which needs to be cross-validated for optimal performance.

## 6 Conclusion

In this paper, we investigate the potentials and limitations of BN in online continual learning, which is a common component in most existing methods but has not been actively studied. We showed that while BN can facilitate knowledge transfer by normalizing along the mini-batch dimension, the cross-task normalization effect hinders older tasks' performance and increases catastrophic forgetting. This limitation motivated us to propose CN, a novel normalization layer especially designed for online continual learning settings. Our extensive experiments corroborate our findings and demonstrate the efficacy of CN over other normalization strategies. Particularly, we showed that CN is a plug-in replacement for BN and can offer significant improvements on different evaluation metrics across different online settings with minimal computational overhead.

## ACKNOWLEDGEMENT

The first author is supported by the SMU PGR scholarship. We thank the anonymous Reviewers for the constructive feedback during the submission of this work.

## ETHIC STATEMENT

No human subjects were involved during the research and developments of this work. All of our experiments were conducted on the standard benchmarks in the lab-based, controlled environment. Thus, due to the abstract nature of this work, it has minimal concerns regarding issues such as discrimination/bias/fairness, privacy, etc.

## REPRODUCIBILITY STATEMENT

In this paper, we conduct the experiments *five* times and report the mean and standard deviation values to alleviate the randomness of the starting seed. In Appendix D, we provide the full details of our experimental settings.

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

## APPENDIX ORGANIZATION

The Appendix of this work is organized as follows. In Appendix A we detail the evaluation metrics used in our experiments. We provide the details of additional normalization layers that we compared in our experiments in Appendix B. In Appendix C, we discuss how our results related to existing studies. Next, Appendix D provides additional details regarding our experiments, including the information of the benchmarks and additional ablation studies. Lastly, we conclude this work with the implementation in Appendix E.

## A   EVALUATION METRICS

For a comprehensive evaluation, we use three standard metrics to measure the model's performance: Average Accuracy (Lopez-Paz & Ranzato, 2017), Forgetting Measure (Chaudhry et al., 2018), and Learning Accuracy (Riemer et al., 2019). Let $a_{i,j}$ be the model's accuracy evaluated on the test set of task $\mathcal{T}_j$ after it finished learning the task $\mathcal{T}_i$. The aforementioned metrics are defined as:

- **Average Accuracy (higher is better):**

$$\text{ACC}(\uparrow) = \frac{1}{T} \sum_{i=1}^{T} a_{T,i}. \tag{7}$$

- **Forgetting Measure (lower is better):**

$$\text{FM}(\downarrow) = \frac{1}{T-1} \sum_{j=1}^{T-1} \max_{l \in \{1,\dots T-1\}} a_{l,j} - a_{T,j}. \tag{8}$$

- **Learning Accuracy (higher is better):**

$$\text{LA}(\uparrow) = \frac{1}{T} \sum_{i=1}^{T} a_{i,i}. \tag{9}$$

The Averaged Accuracy (ACC) measures the model's overall performance across all tasks and is a common metric to compare among different methods. Forgetting Measure (FM) measures the model's forgetting as the averaged performance degrades of old tasks. Finally, Learning Accuracy (LA) measures the model's ability to acquire new knowledge. Note that in the task free setting, the task identifiers are *not* given to the model at any time and are only used to measure the evaluation metrics.

# B  ADDITIONAL NORMALIZATION LAYERS

This section covers the additional normalization layers used in our experiments.

## B.1  LAYER NORMALIZATION (LN)

Layer Normalization Ba et al. (2016) was proposed to address the discrepancy between training and testing in BN by normalization along the spatial dimension of the input feature. Particularly, LN computes the moments to normalize the activation as:

$$\mu_{LN} = \frac{1}{CWH} \sum_{c=1}^{C} \sum_{w=1}^{W} \sum_{h=1}^{H} \boldsymbol{a}_{bcwh}$$

$$\sigma_{LN}^2 = \frac{1}{CWH} \sum_{c=1}^{C} \sum_{w=1}^{W} \sum_{h=1}^{H} (\boldsymbol{a}_{bcwh} - \mu_{LN})^2 \tag{10}$$

## B.2  INSTANCE NORMALIZATION (IN)

Similar to LN, Instance Normalization Ulyanov et al. (2016) also normalizes along the spatial dimension. However, IN normalizes each channel separately instead of jointly as in LN. Specifically, IN computes the moments to normalize as:

$$\mu_{IN} = \frac{1}{WH} \sum_{w=1}^{W} \sum_{h=1}^{H} \boldsymbol{a}_{bcwh}$$

$$\sigma_{IN}^2 = \frac{1}{WH} \sum_{w=1}^{W} \sum_{h=1}^{H} (\boldsymbol{a}_{bcwh} - \mu_{LN})^2 \tag{11}$$

## B.3  BATCH RENORMALIZATION (BRN)

Batch Renormalization Ioffe (2017) was proposed to alleviate the non-i.i.d or small batches in the training data. BRN proposed to normalize training data using the running moments estimated so far by the following re-parameterization trick:

$$\boldsymbol{a}_{BRN} = \boldsymbol{\gamma} \left( r \left( \frac{\boldsymbol{a} - \boldsymbol{\mu}_{BN}}{\boldsymbol{\sigma}_{BN} + \epsilon} \right) + d \right) + \boldsymbol{\beta} \tag{12}$$

where

$$r = \texttt{stop\_grad} \left( \texttt{clip}_{[1/r_{max}, r_{max}]} \left( \frac{\sigma_{BN}}{\sigma_r} \right) \right)$$

$$d = \texttt{stop\_grad} \left( \texttt{clip}_{[-d_{max}, d_{max}]} \left( \frac{\boldsymbol{\mu}_{BN} - \boldsymbol{\mu}_r}{\sigma_r} \right) \right),$$

where $\texttt{stop\_grad}$ denotes a gradient blocking operation where its arguments are treated as constant during training and the gradient is not back-propagated through them.

## B.4  SWITCHABLE NORMALIZATION (SN)

While LN and IN addressed some limitations of BN, their success are quite limited to certain applications. For example, LR is suitable for recurrent neural networks and language applications (Ba et al., 2016; Xu et al., 2019), while IN is widely used in image stylization and image generation (Ulyanov et al., 2016; 2017). Beyond their dedicated applications, BN still performs superior to LR and IN. Therefore, SN (Luo et al., 2018) is designed to take advantage of both LN and IN while enjoying BN's strong performance. Let $\Omega = \{\text{BN,LN,IN}\}$, SN combines the moments of all three normalization layers and performs the following normalization:

$$\boldsymbol{a}' = \boldsymbol{\gamma} \left( \frac{\boldsymbol{a} - \sum_{k \in \Omega} \boldsymbol{w} \boldsymbol{\mu}_k}{\sqrt{\boldsymbol{w}' \boldsymbol{\sigma}_k^2 + \epsilon}} \right) + \boldsymbol{\beta}, \tag{13}$$

Table 7: Dataset summary

|  | #task | img size | #training imgs | #testing imgs | #classes | Section |
|---|---|---|---|---|---|---|
| pMNIST | 5 | 28×28 | 10,000 | 50,000 | 10 | Section 3.2 |
| Split CIFAR10 | 5 | 3×32×32 | 50,000 | 10,000 | 10 | Section 5.2 |
| Split CIFAR100 | 20 | 3×84×84 | 50,000 | 10,000 | 100 | Section 5.1 |
| Split Mni IMN | 20 | 3×84×84 | 50,000 | 10,000 | 100 | Section 5.1 |
| Split Tiny IMN | 10 | 3×64×64 | 100,000 | 10,000 | 200 | Section 5.2 |
| COCOseq | 4 | 3×224×224 | 35,072 | 6,346 | 70 | Section 5.3 |
| NUS-WIDEseq | 6 | 3×224×224 | 48,724 | 2,367 | 49 | Section 5.3 |

where $w$ and $w'$ are the learned blending factors to combine the three moments and are learned by backpropagation.

## C    EXISTING REMEDIES FOR THE CROSS-TASK NORMALIZATION EFFECT

From Table 1, we can see that the performance gap between BN and BN* is much smaller in ER compared to the Single method. Since ER employs an episodic memory, its moments are estimated from both the current data and a small subset of previous memory data. However, there still exists a gap compared to the optimal BN's moments, as shown by large $\Delta_\mu$ and $\Delta_{\sigma^2}$ values. In literature, there exist efforts to such as herding (Rebuffi et al., 2017) and coreset (Nguyen et al., 2018) to maintain a smaller subset of samples that produces the same feature mean (first moment) as the original data. However, empirically they provide insignificant improvements over the random sampling strategies (Javed & Shafait, 2018; Chaudhry et al., 2019a). We can explain this result by the internal covariate shift property in BN: a coreset selected to match the whole dataset's feature mean is corresponding to the network parameters producing such features. After learning a new task, the network parameters are changed to obtain new knowledge and become incompatible with the coreset constructed using the previous parameters. Unfortunately, we cannot fully avoid this negative effect without storing all previous data. With limited episodic memory, lost samples will never contribute to the global moments estimated by the network parameters trained on newer tasks.

## D    ADDITIONAL EXPERIMENTS

### D.1    DATASET SUMMARY

Table 7 summaries the datasets used throughout our experiments.

### D.2    EXPERIMENT SETTINGS

This section details the experiment settings in our work.

**Toy pMNIST Benchmark**    In the toy pMNIST benchmark, we construct a sequence of five task by applying a random but fixed permutation on the original MNIST LeCun et al. (1998) dataset to create a task. Each task consists of 1,000 training samples and 10,000 testing samples as the original MNIST dataset. In the pre-processing step, we normalize the pixel value by dividing its value by 255.0. Both the Single and Experience Replay (ER) methods were trained using the Stochastic Gradient Descent (SGD) optimizer with mini-batch size 10 over one epoch with learning rate 0.03. This benchmark follows the "single-head" setting Chaudhry et al. (2018), thus we only maintain a single classifier for all tasks and task-identifiers are not provided in both training and testing.

**Online Task-IL Continual Learning Benchmarks (Split CIFAR100 and Split Mini IMN)**    In the data pre-processing step, we normalize the pixel density to $[0-1]$ range and resize the image size to $3 \times 32 \times 32$ and $3 \times 84 \times 84$ for the Split CIFAR100 and Split minIMN benchmarks, respectively. No other data pre-processing steps are performed. All methods are optimized by SGD with mini-batch size 10 over one epoch. The episodic memory is implemented as a Ring buffer Chaudhry et al. (2019a) with 50 samples per task.

Table 8: Evaluation metrics of CN variants on the Split Mini IMN and COCOseq benchmarks (overall classes). We report the mean results over five runs. Bold indicates highest score, "tw" indicates tied weight in the affine parameters

| CN Variant | Split Mini IMN | | | COCOseq | | |
|---|---|---|---|---|---|---|
| | ACC | FM | LA | C-F1 | O-F1 | mAP |
| GN→BN(original) | **64.28** | **8.08** | **70.90** | **53.1** | **49.8** | **55.1** |
| BN→GN | 62.63 | 9.92 | 71.43 | 51.5 | 47.9 | 53.8 |
| GN→BN+tw | 62.17 | 9.21 | 70.17 | 51.1 | 47.7 | 53.3 |
| BN→GN+tw | 62.23 | 8.20 | 69.30 | 50.0 | 47.4 | 52.5 |

**Online Task-IL and Class-IL Continual Learning Benchmarks (Split CIFAR-10 and Split Tiny IMN)**   We follow the protocol as the DER++ work (Buzzega et al., 2020) except the number of epochs, which we set to one. Particularly, training is performed with data augmentation on both the data stream and the memory samples (random crops, horizontal flips). We follow the configurations provided by the authors, including the hyper-parameters of mini-batch size, the reservoir sampling memory management strategy, learning rates, etc. For the memory size of 2560, which were not conducted in the original paper, we use the same setting as the case of 5120 memory slots.

**Online Task-Free Long-Tailed Continual Learning (COCOseq and NUS-WIDEseq)**   We follow the same settings as the original work Kim et al. (2020) in our experiments. Particularly, we train each method using the Adam optimizer Kingma & Ba (2014) with default hyper-parameters ($\beta_1 = 0.9, \beta_2 = 0.999, \epsilon = 1e-4$), mini-batch size 10 over one epoch. The episodic memory is implemented as a reservoir buffer using the PRS (Kim et al., 2020) strategy with total 2,000 slots.

### D.3   COMPARING AMONG CN VARIANTS

Recall that our CN design starts with GN without the affine parameters and follows up with BN. We consider an alternative setting of CN, denoted by BN-GN, by performing BN and then GN, in which the affine parameters are added at the GN layer. For each setting, we create a "tied-weight" variant where the affine parameters are also used in the first normalization step. We consider the split Mini IMN with ER and the COCOseq benchmark with PRS and report the result in Table 8. Due to space constraints, we only report the mean results over five runs. For the COCOseq benchmark, we only report the metrics in the Overall category. All variants are performing comparable to our original design on the Split Mini IMN benchmarks in terms of ACC. However, they have higher FM, indicating that normalizing features before BN is beneficial and sharing weights in the first normalizing steps is not helpful. However, on the challenging COCOseq benchmark, CN's original design performs much better than other variants across all metrics. Moreover, adding the affine parameters to the first normalization step (tied weight) reduces the normalization properties of the features and reduces overall performance.

### D.4   CHANGING RATE OF THE MOMENTS IN BATCH NORMALIZATION

We now investigate the changing rate of the moments in BN. We consider the toy MNIST experiment as introduced in Section 3.2 with the naive finetuning model (Single). Throughout training, we measure the difference between the running moments of BN and the optimal moments from BN* estimated at the end of each task. Figure 3 plots the changing of each moment from the two-layers MLP backbone throughout the learning of five pMNIST tasks. Notably, only the mean of the second layer remains stable throughout learning. All the other moments, especially from the first layer, quickly diverge as the model observes more tasks. Moreover, the discrepancy in the moments of the first layer causes a difference between the first hidden features, which cascades through the network depth and results in inaccurate final prediction in BN compared to BN*.

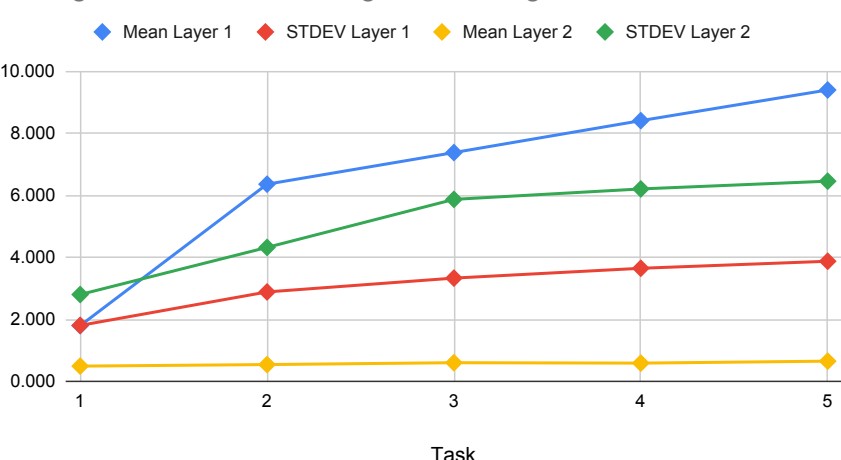

Figure 3: Changes of moments (L1 distance - y axis) between BN and BN* throughout training on a 2-layers MLP backbone. Standard deviation values are omitted due to small values (e.g. $< 0.1$)

Table 9: Comparison with BN* on the 4-tasks Split CIFAR-100 benchmarks. Bold indicates the best averaged scores

| ER | Split CIFAR-100 (4 tasks) | | |
|---|---|---|---|
| | ACC | FM | LA |
| BN | 60.14±3.47 | 6.21±1.99 | 63.98±2.38 |
| BN* | 61.38±2.46 | 5.87±1.37 | **65.01±1.78** |
| GN | 55.96±2.43 | **2.23±0.79** | 53.25±2.44 |
| CN | **62.18±0.56** | 5.66±0.76 | 64.94±1.68 |

### D.5 ADDITIONAL RESULTS OF BN*

We further investigate the performance of BN* on the Split-CIFAR100 benchmark and report and report the result in Table 9. Since BN* requires calculating the true moments from all observed data, we cannot scale it to the standard benchmark of 17 tasks. Table 9 only consider a sequence of four tasks, which is the maximum memory capacity that our GPU allows. Interestingly, we observe that GN achieves low FM in this experiment, suggesting that GN does not suffer much from catastrophic forgetting early on during training, or when the sequence of tasks is short. As there are more tasks, we expect the gap among different methods to be more significant as demonstrated throughout our work.

### D.6 DECOUPLING THE BENEFITS OF EXPERIENCE REPLAY FROM CN

We investigate the benefit of CN while decoupling from the episodic memory for experience replay by considering the naive finetuning method (Single) with different normalization layers and report the results in Table 10.

On both Split CIFAR-100 and Split mini IMN benchmarks, we observe consistent improvements of CN over BN in the final performance (1% on Split CIFAR-100 and 2% on Split mini IMN). This improvement comes from the reduction of catastrophic forgetting (around 0.7% reduction in FM) and the improvements of forward transfer (from 1 to 2% increases in LA). We also observe that GN achieved competitive performance on the Split CIFAR-100 benchmark thanks to its ability to alleviate the cross-task normalization effects compared to other methods that have a BN component.

Table 10: Evaluation metrics of the naive single strategy on the task-aware Split CIFAR-100 and Split mini IMN benchmarks. Bold indicates the best averaged scores

| Single | Split CIFAR-100 | | | Split mini IMN | | |
|---|---|---|---|---|---|---|
| | ACC | FM | LA | ACC | FM | LA |
| BN | 33.47±2.24 | 33.93±2.23 | 65.74±2.22 | 32.26±2.42 | 29.76±0.98 | 57.20±1.25 |
| BRN | 28.74±2.53 | 31.78±3.06 | 58.02±1.94 | 25.60±0.93 | 25.77±2.04 | 49.43±1.37 |
| GN | **36.66±2.32** | **17.40±2.33** | 52.62±1.61 | 28.05±1.17 | **12.97±2.02** | 39.60±1.55 |
| CN | 34.52±2.62 | 33.27±1.23 | **66.37±1.00** | **34.23±1.37** | 29.05±0.74 | **61.11±1.61** |

Table 11: Evaluation metrics of DER++ with different normalization layers on the Split CIFAR-10 and Split Tiny IMN benchmarks. Parentheses indicates the number of groups in CN. Bold indicates the best averaged scores. This is the full version of Table 4

| Buffer | Method | Split CIFAR-10 | | | | Split Tiny IMN | | | |
|---|---|---|---|---|---|---|---|---|---|
| | | Class-IL | | Task-IL | | Class-IL | | Task-IL | |
| | | ACC(↑) | FM(↓) | ACC(↑) | FM(↓) | ACC(↑) | FM(↓) | ACC(↑) | FM(↓) |
| 500 | BN | 48.9±4.5 | 33.6±5.8 | 82.6±2.3 | 3.2±1.9 | 6.7±0.1 | 38.1±0.5 | 40.2±0.9 | 6.5±1.0 |
| | **IN** | 35.9±2.0 | 45.0±3.6 | 78.8±1.6 | 1.8±0.9 | 2.2±0.5 | **16.9±0.9** | 20.6±0.5 | **1.7±0.5** |
| | GN | 46.8±5.1 | 29.7±11.5 | 82.1±5.2 | **1.9±1.2** | 7.2±0.4 | 36.0±2.4 | 41.1±0.5 | 4.5±1.8 |
| | SN | 42.3±3.9 | 32.3±11.6 | 80.6±3.4 | 2.6±2.0 | 4.5±1.0 | 25.0±2.4 | 33.6±1.3 | 2.5±1.9 |
| | CN (G=8) | 48.9±0.3 | **27.9±5.0** | 84.7±0.5 | 2.2±0.3 | **7.3±0.9** | 37.5±2.3 | **42.2±2.1** | 4.9±2.2 |
| | CN (G=32) | **51.7±1.9** | 28.2±4.0 | **86.2±2.2** | 2.0±1.3 | 6.5±0.7 | 40.1±1.7 | 40.6±1.4 | 6.8±1.9 |
| 2560 | BN | 52.3±4.6 | 29.7±6.1 | 86.6±1.6 | 0.9±0.7 | 11.2±2.3 | 36.0±2.0 | 50.8±1.7 | 2.8±1.2 |
| | IN | 36.1±2.8 | 37.8±8.8 | 79.1±1.1 | 0.8±1.4 | 2.4±0.4 | **19.2±1.5** | 25.4±1.7 | **0.6±0.5** |
| | GN | 53.3±3.5 | 28.8±3.2 | 87.3±2.4 | **0.6±0.7** | 11.0±1.3 | 35.5±1.2 | 50.9±3.0 | 1.9±1.6 |
| | SN | 42.3±3.9 | 30.8±10.4 | 80.0±6.4 | 4.7±5.2 | 4.4±1.4 | 27.3±2.9 | 37.6±3.0 | 3.5±1.8 |
| | CN (G=8) | 53.7±3.4 | 25.5±4.7 | 87.3±2.7 | 1.6±1.6 | 10.7±1.3 | 38.0±1.2 | **52.6±1.2** | 1.5±0.5 |
| | CN (G=32) | **57.3±2.0** | **21.6±5.6** | **88.4±1.1** | 1.7±1.1 | **11.9±0.3** | 36.7±1.2 | 51.5±0.2 | 2.8±0.9 |
| 5120 | BN | 52.0±7.8 | 26.7±10.3 | 85.6±3.3 | 2.0±1.5 | 11.2±2.7 | 36.8±2.0 | 52.2±1.7 | 2.6±1.5 |
| | IN | 32.2±4.5 | 41.5±3.6 | 77.2±3.8 | 2.1±2.3 | 2.3±0.6 | **18.0±1.3** | 25.5±2.0 | 1.6±0.9 |
| | GN | 48.6±2.6 | 32.2±4.8 | 86.5±0.8 | **0.5±0.5** | 9.2±1.8 | 36.8±1.6 | 48.0±6.0 | 4.9±6.3 |
| | SN | 42.9±8.9 | 36.2±8.3 | 81.1±2.4 | 3.6±2.4 | 4.3±0.6 | 23.3±1.9 | 37.9±3.0 | 2.7±1.9 |
| | CN (G=8) | 54.1±4.0 | 24.0±4.0 | 87.1±2.8 | 0.7±0.7 | **12.2±0.6** | 34.6±2.6 | 53.1±1.8 | 3.1±1.9 |
| | CN (G=32) | **57.9±4.1** | **22.2±1.0** | **88.3±0.9** | 1.3±0.9 | **12.2±0.2** | 35.6±1.3 | **54.9±1.5** | 1.5±1.1 |

However, GN's performance falls short on the more challenging Split mini IMN benchmark because it cannot learn individual task well.

Please note that this naive strategy is extremely challenging to train in continual learning because it does not have any mechanism to prevent catastrophic forgetting or support forward transfer. The difference in the first layer normalization layer could cascade through the network depth and result in a huge difference in the final prediction when using BN versus BN*.

## D.7 ADDITIONAL RESULTS OF DER++

In Table 11 we results of additional normalization layers of DER++ on the Split CIFAR-10 and Split Tiny IMN benchmarks. This is the full version of the results presented in Section 5.2 and Table 4. Overall, the results are consistent with the current ones in that GN and SN are generally more resistant to catastrophic forgetting (lower FM), but they lack the ability to forward transfer knowledge, resulting in lower overall performance (ACC) compared to our CN, and sometimes BN. The only notable difference is that IN achieved the lowest FM in the Class-IL Tiny IMN benchmark. By inspecting the results, we found that since this benchmark is more challenging, IN struggled to learn individual tasks and only achieved slightly better final ACC than random guessing. Therefore, IN had significantly lower FM because it did not accumulate much knowledge during training, which is similar to the no-normalization method.

Table 12: Effects of different moving average strategies on the Split CIFAR-100 benchmark, with ER+BN

| Moving Average | | Split CIFAR-100 | | |
|---|---|---|---|---|
| | | ACC | FM | LA |
| CMA | N/A | 35.86±0.90 | 20.24±1.30 | 45.53±3.65 |
| | $\eta = 0.01$ | 62.90±2.03 | 6.32±2.32 | 64.81±1.15 |
| | $\eta = 0.05$ | 63.61±1.11 | 7.18±1.08 | 68.43±0.87 |
| EMA | $\eta = 0.1$ | 64.97±1.09 | 9.24±1.98 | 71.56±0.75 |
| | $\eta = 0.5$ | 62.43±2.65 | 8.54±2.76 | 68.86±0.51 |
| | $\eta = 0.9$ | 61.46±2.36 | 9.03±2.38 | 66.81±1.49 |

## D.8 ANALYSIS OF THE MOVING AVERAGE IN BN

We now explore how the moving average affects BN. First, we consider the cumulative moving average (CMA) strategy which keeps the standard average of the statistics for the first $n$ mini-batches of data as:

$$\boldsymbol{x}_{\text{CMA}} = \frac{\boldsymbol{x}_1 + \boldsymbol{x}_2 + \ldots + \boldsymbol{x}_n}{n}, \tag{14}$$

where $\boldsymbol{x}_{\text{CMA}}$ is the CMA moving average statistic, $\boldsymbol{x}_t$ is the $t-$ mini-batch statistic.

Next, we consider the standard exponential moving average strategy (EMA). For consistency with the Pytorch framework, we refer to the update of the running statistics in BN as

$$\hat{\boldsymbol{x}}_{\text{new}} = (1 - \eta) \times \hat{\boldsymbol{x}} + \eta \times \boldsymbol{x}_t, \tag{15}$$

where $\hat{\boldsymbol{x}}$ is the current running statistic, and $\boldsymbol{x}_t$ is the current mini-batch statistic. WIth this formulation, lower $\eta$ values assign higher weights to older statistics, while higher $\eta$ values will focus more on recent statistics.

We report the results of CMA versus different EMA's $\eta$ configurations on the Split CIFAR-100 benchmark with ER under the task-aware setting in Table 12. From the results, we draw the following conclusions:

- The standard CMA strategy yields undesirable performance and is worse than EMA. The reason is that when using CMA, the network parameters at the early stages of training contribute equally to CMA moments compared to the latest parameter, which is problematic because the early parameters are not well-trained and do not observe the newer tasks' samples. When using the exponential running average (the standard practice), the contributions of early parameters to the running moments degrades exponentially compared to the recent, well-trained parameters.

- For BN with EMA, low values of $\eta = 0.01$ or $\eta = 0.05$ slow the running statistics' changing rate, which makes the EMA statistics more stable and results in less forgetting (indicated by lower FM). However, these settings cannot take advantage of the statistics from the recent model's parameters and result in lower forward transfer (lower LA), which results in worse overall performance (lower ACC).

- On the other hand, with $\eta > 0.1$, the running statistics can be too sensitive to the noisy updates from mini-batches, resulting in worse performance than the standard setting of $\eta = 0.1$

It is also worth noting that any improvements with the EMA moving average can also benefit our CN via adapting its BN component accordingly. It is also interesting to note that our BN* strategy is equivalent to not estimating a running averaged but calculating these statistics at the time of evaluation using the most recent model.

## D.9 STANDARD DEVIATIONS OF THE COCOSEQ AND NUS-WIDESEQ BENCHMARKS

This section provides the standard deviation (stdev) of results for the long-tailed online continual learning experiments with the COCOseq and NUS-WIDEseq benchmarks in Section 5.3. Table 13

Table 13: Standard deviations of the evaluation metrics of the PRS strategy on the COCOseq and NUS-WIDEseq benchmarks

| COCOseq | Majority | | | Moderate | | | Minority | | | Overall | | |
|---|---|---|---|---|---|---|---|---|---|---|---|---|
| | C-F1 | O-F1 | mAP | C-F1 | O-F1 | mAP | C-F1 | O-F1 | mAP | C-F1 | O-F1 | mAP |
| BN | 0.77 | 1.33 | 1.04 | 0.98 | 1.28 | 1.04 | 2.35 | 2.33 | 1.78 | 0.89 | 1.24 | 1.07 |
| CN(ours) | 1.20 | 1.97 | 1.12 | 0.61 | 1.27 | 0.66 | 1.73 | 1.53 | 0.59 | 0.64 | 1.22 | 0.63 |
| NUS-WIDEseq | Majority | | | Moderate | | | Minority | | | Overall | | |
| | C-F1 | O-F1 | mAP | C-F1 | O-F1 | mAP | C-F1 | O-F1 | mAP | C-F1 | O-F1 | mAP |
| BN | 1.01 | 1.97 | 0.91 | 1.72 | 1.77 | 0.68 | 1.95 | 1.78 | 1.44 | 0.47 | 0.27 | 0.55 |
| CN(ours) | 2.38 | 1.59 | 1.33 | 2.78 | 3.06 | 0.97 | 1.72 | 1.05 | 0.58 | 1.26 | 1.40 | 0.34 |

Table 14: Standard deviations of the forgetting measure (FM($\downarrow$) ) of each metric from the PRS strategy on the COCOseq and NUS-WIDEseq benchmarks, lower is better

| COCOseq | Majority | | | Moderate | | | Minority | | | Overall | | |
|---|---|---|---|---|---|---|---|---|---|---|---|---|
| | C-F1 | O-F1 | mAP | C-F1 | O-F1 | mAP | C-F1 | O-F1 | mAP | C-F1 | O-F1 | mAP |
| BN | 3.77 | 4.60 | 0.44 | 3.11 | 2.80 | 1.63 | 4.63 | 4.32 | 2.34 | 2.63 | 2.93 | 1.29 |
| CN(ours) | 3.31 | 3.62 | 2.53 | 1.46 | 1.54 | 1.72 | 5.72 | 6.32 | 0.80 | 1.26 | 1.34 | 1.34 |
| NUS-WIDEseq | Majority | | | Moderate | | | Minority | | | Overall | | |
| | C-F1 | O-F1 | mAP | C-F1 | O-F1 | mAP | C-F1 | O-F1 | mAP | C-F1 | O-F1 | mAP |
| BN | 7.35 | 8.19 | 1.49 | 6.60 | 7.03 | 1.35 | 13.34 | 13.24 | 4.14 | 4.71 | 5.11 | 1.08 |
| CN(ours) | 7.59 | 6.60 | 4.16 | 8.89 | 9.14 | 3.77 | 7.35 | 7.15 | 7.42 | 2.00 | 2.38 | 3.08 |

shows the stdev values of the evaluation metrics while Table 14 shows the stdev values of the FM associating with each evaluation metric.

In most cases, our CN has comparable stdev values with BN on both benchmarks. While these values are relatively small on COCOseq (mostly smaller than 4%), the stdev values are higher on the NUS-WIDEseq benchmark, especially for the FM. This phenomenon could happen because the NUS-WIDE dataset is highly noisy, which greatly impacts the continual learning benchmark. Nevertheless, both BN and our CN have similar FM values for most evaluation metrics in this scenario.

## E    CONTINUAL NORMALIZATION IMPLEMENTATION

In the following, we provide the CN's implementation based on Pytorch (Paszke et al., 2017).

```
class CN(_BatchNorm):
    def __init__(self, num_features, eps = 1e-5, G = 32, momentum):
        super(_CN, self).__init__(num_features, eps, momentum)
        self.G = G

    def forward(self, input):
        out_gn = F.group_norm(input, self.G, None, None, self.eps)
        out = F.batch_norm(out_gn, self.running_mean, self.running_var,
            self.weight, self.bias,
            self.training, self.momentum, self.eps)
        return out
```

