# OpenReview forum: "Continual Normalization: Rethinking Batch Normalization for Online Continual Learning"
_ICLR.cc/2022/Conference — ICLR 2022 Poster_

### Official Review · Reviewer_9JHA · 2021-10-31

**Correctness:** 4
**Technical Novelty And Significance:** 3
**Empirical Novelty And Significance:** 3
**Recommendation:** 6
**Confidence:** 4

**Main Review:**

Strong Points

* The paper takes one of the most import issues in continual learning: non-stationary online CL setting. For me, the problem itself is real and practical.

* The proposed approach is reasonable and addresses the problem of cross-task normalization in CL setting.

* The proposed adaptive normalization layers can be a plug-in replacement of BN.

* The authors made extensive comparison between CN and prior work such as BN, IN, and GN.

* Overall, the paper is well written. In particular, the Related Work section has a nice flow and puts the proposed method into context. Despite the method having limited novelty, the method has been well motivated by pointing out the limitations in SOTA methods.

* The authors provide code for reproducing the results in the paper.

Weak Points

* The proposed adaptive normalization layer is a straight-forward combination of spatial and batch normalization. So the novelty is limited.

* Table 2, in DER++ setting, CN's FM score (for catastrophic forgetting) is not as good as GN's. Please explain the reason for that.


**Summary Of The Paper:**

This paper investigates the problem of cross-task normalization in a non-stationary online continual learning(CL) setting. It argues that batch normalization (BN) is important to CL; however, BN in the current form introduces a bias towards current task, leading to catastrophic forgetting. The paper presents a continual normalization(CN) layer to reduce this bias. In particular, it combines spatial and batch normalization into one layer that is suitable for CL. The authors conducted experiments to evaluate the performance of the proposed method.


**Summary Of The Review:**

Overall, I vote for marginally accepting. I like the idea of cross-task normalization and handling it by the proposed adaptive normalization layer. My major concern is about the limited novelty of the paper and the performance of CN on catastrophic forgetting (see weakness above). Hopefully the authors can address my concern in the rebuttal period.

[After rebuttal]
After reading the other reviewers' comments and the authors' rebuttal, I confirm my rating.

---

> ### Author Response · Authors · 2021-11-18
> **Response to Reviewer 9JHA**
>
> **Update 1 (Nov 26)** We have added Section 6.4 in the main paper to discuss the limitations of our work as pointed out by R 9jHA and R BLEs.
>
> **Concern #1: limited novelty**
>
> We believe that CN’s simplicity is one of its strengths, allowing it to seamlessly replace BN for continual learning without much additional effort. In addition, one important contribution of our study is investigating the benefits and drawbacks of different normalization layers to the online continual learning setting.
>
> **Concern #2: CN’s FM score is not as good as GN for DER++ in Table 2**
>
> DER++ introduces an additional regularization based on soft labels to further alleviate catastrophic forgetting. We believe that this component allows GN to achieve lower FM for DER++ compared to CN because it over-emphasizes on reducing FM while CN has to balance between reducing FM and improving LA.

---

> > ### Author Response · Authors · 2021-11-26
> > **Anything else the Reviewer would like us to respond to?**
> >
> > Dear Reviewer,
> >
> > As the discussion period is closing soon, we would like to check with the Reviewer if our reply has addressed your concerns. Additionally, if you still have any further questions, please let us know and we'd be happy to take a look.
> > Thank you.

---

> > > ### Comment · Reviewer_9JHA · 2021-11-30
> > > **Response to authors**
> > >
> > > Thanks for the detailed answer. My concerns are addressed. After reading the other reviews and the answers, I confirm my rating.

---

### Official Review · Reviewer_9jXz · 2021-11-02

**Correctness:** 3
**Technical Novelty And Significance:** 3
**Empirical Novelty And Significance:** 2
**Recommendation:** 5
**Confidence:** 4

**Main Review:**

Strengths:
1. The paper is well-written and easy to understand.
2. The idea of Continual Normalization is novel under the continual learning setting.
3. Multiple continual learning settings, including task-incremental, class-incremental and long-tailed continual learning, make the experiments part comprehensive.


Weaknesses and Questions:
1. Section 2 is weird, maybe section 3 should be section 2.1?
2. In (6), the Batch Norm operation still exists, so the cross task normalization effect is not directly addressed. The intuition of why an "adaptive normalization" at test time could reduce forgetting (or the cross task normalization effect) should be elaborated better.
3. It's nice to see the method worked with replay based method (Table 2, 4). However, I still wonder the performance of CN when there is no replay involved. As replay in some sense already alleviates the cross-task normalization effect, it would be better to decouple the influence of CN and replay.
4. (minor) According to the paper, the method can combine the merits of both BN and GN under the continual learning setting. But the method itself is not limited to continual learning problem. Any intuition about how CN would perform under the i.i.d. supervised learning setting?

**Summary Of The Paper:**

This paper proposes a novel normalization layer called Continual Normalization for continual learning. In the paper, the authors point out the problem of global moment bias of Batch Norm. To address the problem, the authors combine the advantages of Group Norm and Batch Norm together to improve sharing while reduce forgetting. Comprehensive experiments on various benchmarks and continual settings demonstrate the effectiveness of the proposed method.

**Summary Of The Review:**

The proposed Continual Normalization is novel, however, the reasoning and intuition behind the idea should be elaborated better. Moreover, additional experiments without replay-based method can better demonstrate the effectiveness of the method.

---

> ### Author Response · Authors · 2021-11-18
> **Response to Reviewer 9jXz**
>
> **Concern #2: why an adaptive normalization layer at test time could reduce forgetting should be elaborate better**
>
> Please refer to our respond to Concern #1b from R BLEs for our answer.
>
> **Update 1 (Nov 26)**  We refer the Reviewer to our discussion with R BLEs, concern #1b and the follow up discussion for our answer. For a more detailed discussion regarding how the empirical results support our argument, please refer to our response to concern #3 of R 8iRV
>
> **Concern #3: decouple CN from replay**
>
> Please refer to our response to Concern #2 from R 8iRv for the results and discussion.
>
> **Concern #4: CN under the standard i.i.d supervised learning setting**
>
> We develop CN to address the BN’s weakness of in-accurate estimates of the running moments. Therefore, we suspect that CN can be beneficial in the scenarios of small batch sizes, or in the presence of data imbalance. However, exploring such settings is beyond the scope of our study.

---

> > ### Author Response · Authors · 2021-11-26
> > **Anything else the Reviewer would like us to respond to?**
> >
> > Dear Reviewer,
> >
> > As the discussion period is closing soon, we would like to check with the Reviewer if our reply has addressed your concerns. Additionally, if you still have any further questions, please let us know and we'd be happy to take a look.
> > Thank you.

---

### Official Review · Reviewer_BLEs · 2021-11-02

**Correctness:** 3
**Technical Novelty And Significance:** 3
**Empirical Novelty And Significance:** 4
**Recommendation:** 8
**Confidence:** 4

**Main Review:**

Strengths:
- The paper focuses on an important problem of activation normalization which to my knowledge has not yet been properly explored in the continual learning setting. While a lot of effort has been directed towards proposing better algorithms on top of existing network architectures, not that much research has been done in the direction of fixing the architectures themselves and in particular fixing specific parts of the architectures, such as the normalization schemes.
- The practicality and relevance of the proposed technique are high. Continual Normalization is very easy to use and adapt to existing architectures, which makes the paper useful for the community.
- The improvements offered by the method are mostly consistent between different tested settings, with few exceptions.
- The empirical evaluation provided in the paper is extensive and authors consider multiple important perspectives, such as the computational complexity of CN (the running time in Table 3), how it behaves in different CL settings (Task-IL, Class IL), and how an "oracle method" performs (BN*).

Weaknesses:
- Although the paper provides an empirical investigation of the issue of normalization in a continual learning setting, it would benefit from a more in-depth study of how different normalization schemes affect the results. Currently, the paper mostly shows that the proposed method works in practice and offers some intuitions to justify their good performance (BN being useful for forward transfer, GN being useful for reducing forgetting), but in my opinion, there is much more to understand here. Why is BN so useful for forward transfer? Why do the "forgetful" properties of BN vanish after applying GN? How fast do the BN statistics change after each task? I feel like questions like those have not yet been answered satisfactorily.
- Although I find the empirical evaluation overall good, there are several points that could be improved or discussed more thoroughly.
  - I think the "oracle" baseline from Table 1, which re-calculates statistics from all training data, is very interesting. However, it is not used in the subsequent experiments, where, in my opinion, it could be helpful in understanding the bigger picture. Additionally, I wanted to ask - do you consider this baseline to be the upper bound of what can be achieved with normalization?
  - In Table 3 you consider many different normalization schemes, but after that, you only show results with BN and CN. I think that understanding how different normalization schemes impact the results in continual learning is an important part of this work and should be included in more experiments (e.g. by putting the extended versions of the results in the appendix).
  - I don't think the paper properly acknowledges certain shortcomings of CN in the experimental section. I think that the overall results are satisfactory, but at the same time, there should be a discussion of the limitations of the method. The paper doesn't really discuss the fact that CN's results are worse for the minority classes of the NUS-WIDEseq dataset, or the fact that sometimes the difference of the mean result of CN and BN is not much higher than the difference of standard deviation, posing a question about the significance of the results. To reiterate, I don't think the results overall are bad, but I would suggest discussing the limitations more openly.
  - The paper does not provide the standard deviation of the results for the case of the COCOSeq benchmark. The authors write that "Due to space constraints, we only report the mean results over five runs.", but as the appendix does not have space constraints, it would be useful to add this information.
  - The paper doesn't consider the impact of the hyperparameter $\eta$ from Equation (3), which controls how fast the running statistics are changing. Since it directly impacts the moments, it possibly could have a high impact on the cross-task normalization effect and the results in the CL settings. Unfortunately, this is not discussed. In fact, if I'm not mistaken, the paper doesn't even state what value of $\eta$ is used in the experiments. Another possible option would be to consider calculating a standard mean instead of a running mean.

Minor comments:
- Section 2 should probably be removed or merged with Section 3 (only two sentences, referring to contents of Section 3)
- Section 4, second sentence: "For convenient" -> "For convenience"
- It would be interesting to consider the non-online continual learning setting (i.e. multiple epochs).

**Summary Of The Paper:**

The paper considers the problem of activation normalization in neural networks in the context of training on non-i.i.d. data in continual learning. The authors showcase the issue of the "cross-task normalization effect", where the data from a given task is normalized by statistics biased towards another (latest) task. Based on this phenomenon, they show that commonly used normalization schemes, such as Batch Normalization or Group Normalization, suffer from certain problems in this setting (high forgetting, low transfer). They propose the Continual Normalization which combines Group Normalization with Batch Normalization to obtain better results and show the improvements empirically on multiple CL datasets.

**Summary Of The Review:**

The research questions asked in this paper are very interesting and go in an orthogonal direction to most of the CL research, which could be beneficial for the CL community. The proposed solution was evaluated empirically with satisfactory results and it's very easy to implement, making it a useful tool for CL researchers and practitioners. On the other hand, the paper would benefit from a more thorough investigation of the problem and fixing certain issues in terms of empirical evaluation. As such, I would say that at the moment it is marginally above the acceptance threshold.

**EDIT after the discussion period:** During the rebuttal process the authors provided important clarifications and introduced significant improvements to the paper. As such, I have decided to increase my score to 8 (accept, good paper).

---

> ### Author Response · Authors · 2021-11-18
> **Response to Reviewer BLEs (part 1)**
>
> **Concern #1a: in-depth study how different normalization schemes affect the results**
>
> Please refer to our response to Concern #3 of R 8iRV for the discussion.
>
> **Concern #1b: Why do the “forgetful” properties of BN vanish after applying GN?**
>
> In our analysis, we have identified that using a set of biased running moments to normalize testing data can cause additional catastrophic forgetting. The adaptive normalization component in CN allows each sample to be normalized differently during testing by allowing each sample to contribute to its own normalizing moments. We believe this design bridges the gap between GN and BN and allow CN to alleviate the negative effect of BN while maintaining its useful property of improving forward transfer.
>
> **Concern #1c: How fast do the BN statistics change after each task?**
>
> Thanks for the interesting suggestion. We examine the BN statistics change by calculating the L1 difference between running moments with the optimal one from BN* at the end of each task. We consider the pMNIST benchmark with a two-layers MLP backbone and plot the difference of each moment in Figure 3, Appendix E.4. Notably, only the mean of the second layer remains stable throughout learning. All the other moments, especially from the first layer, quickly diverge as the model observes more tasks. Moreover, the discrepancy in the moments of the first layer causes a difference between the first hidden features, which cascades through the network depth and results in inaccurate final prediction in BN compared to BN*.
>
> **Concern #2: BN\* in later experiments and BN\* as the upper bound.**
>
> **Concern #2a: BN\* in later experiments.**
>
> Thanks for the suggestion, we have included BN* on the Split-CIFAR 100 benchmark in Table 9, Appendix E.5. Due to the high memory cost of this strategy (BN* requires performing forward calculation on all stored images), we couldn’t run it in the standard 17 tasks as reported in Table 2. Instead, we only report the results up to 4 tasks, which is the maximum capacity that our GPU allows.
> Interestingly, we observe that GN achieves low FM in this experiment, suggesting that GN does not suffer much from catastrophic forgetting early on during training. As there are more tasks, we expect the gap among different methods to be more significant as demonstrated in the remaining experiments.
>
> **Concern #2b: BN\* as the upper bound of the normalization layers.**
>
> Since BN and spatial normalization layers perform vastly different calculations, it is difficult to argue a unified upper bound of all layers. Instead, literature has shown that each type of layer excels in certain applications and situations.
> For BN in the continual learning with image classification domain, we argue that the true upper bound for BN is a variant that uses the true, unbiased global moments to normalize during training and testing. This variant requires re-calculating the global moments using all tasks data at each optimization step, which is computationally prohibitive. In this work, we consider BN* as a simplified version of this upper bound where the global moments are only used during testing.
> Lastly,  in the experiments, we considered Batch Renormalization (BRN), which tries to normalize using the global moments estimated from data stored in the memory. However, BRN does not achieve promising results because it does not have access to old tasks’ data, making its global moments estimated from imbalance samples.
>
>
> **Concern #3: experiments showing the impacts of different normalization methods for DER++.**
>
> Thanks for the suggestion. We have included more normalization layers for the TaskIL and Class IL settings with DER++ in Table 11, Appendix E.7, where we included IN, GN, and SN. Overall, the results are consistent with the current ones in that GN and SN are generally more resistant to catastrophic forgetting than BN (lower FM), but they lack the ability to forward transfer knowledge, resulting in lower overall performance (ACC) compared to our CN, and sometimes BN. The only notable difference is that IN achieved the lowest FM in the Class-IL Tiny IMN benchmark. By inspecting the results, we found that since this benchmark is more challenging, IN struggled to learn individual tasks and only achieved slightly better final results than random guessing. Consequently, IN had significantly lower FM because it didn’t accumulate much knowledge during training and therefore has lower absolute FM, similar to when using no normalization layers in Table 2. However, this may not be a positive result for IN if we compare the relative changes of ACC and FM. For example, in the CLass-IL setting of the Split tiny IMN benchmark with 5120 memory slots, BN has five times higher ACC (11.2 vs 2.3) but its FM is only twice as high as IN's (36.8 vs 18.0).

---

> > ### Author Response · Authors · 2021-11-18
> > **Response to Reviewer BLEs (part 2)**
> >
> > **Concern #4: acknowledge the limitations of the method.**
> >
> > Thanks for the suggestion. We have amended the discussion of Section 6.3 to acknowledge the limitations of our work (text in blue). One possible reason for this discrepancy is the noisy nature of the NUS-WIDE dataset, including the background diversity and huge number of labels per image, which could highly impact the performance of moderate and tail classes.
> >
> > **Concern #5: standard deviations for the COCOseq and NUS-WIDEseq benchmarks.**
> >
> > We provide the standard deviation values (stdev) for the COCOseq and NUS-WIDEseq benchmarks in Table 12, 13 in Appendix E.8.
> > Generally, our CN has comparable stdev values with BN on both benchmarks. While these values are relatively small on COCOseq (mostly smaller than 4\%), they are higher on the NUS-WIDEseq benchmark, especially for the FM. Nevertheless, both BN and CN have similar FM values for most evaluation metrics and their FM in this scenario.
> >
> > **Concern #6: impact of hyper-parameters in Equation (3)**
> >
> > **Concern #6a: impact of $\eta$**
> >
> > For now, all of our experiments with different normalization layers use the same configuration of $\eta = 0.1$, which is the default value in pytorch.
> >
> > **Concern #6b: calculating the standard mean instead of a running mean**
> >
> > Thanks for the suggestion. One important and desirable property of BN is that it must make deterministic predictions at test time, i.e. the prediction of one sample is the same when it is mixed with different mini-batches. Therefore, the standard practice of BN is using standard (mini-batch) moments to normalize the training data while normalizing testing data with running moments to ensure a deterministic behaviour. Using standard moments at test time breaks this property by causing a dependency between the prediction of one sample with other samples in the test mini-batch.
> >
> > **Minors**
> >
> > **Typos in Section 4:** thanks for spotting the typo, we have fixed it in the revision.
> >
> > **The non-online continual learning settings.**
> >
> > Thanks for the suggestions. However, due to the extensive computing requirements of such experiments, we will leave this investigation for future work.

---

> > > ### Comment · Reviewer_BLEs · 2021-11-20
> > > **Response to the Authors**
> > >
> > > I would like to thank the authors for the thorough answer and the new experiments. I appreciate the additional results and I'm considering increasing the score, although there are some issues that still seem problematic to me.
> > >
> > > My major reservation concerns the presentation of the results. The standard deviations in the results seem to be quite high which sometimes can make it unclear whether the improvements are statistically significant. The authors do not address this in their discussion on limitations (Concern #4), so I would like to reiterate this point.
> > >
> > > In particular, the strategy for bolding entries in the tables in the paper does not seem consistent. For example, in Table 10, accuracy for Split Cifar-100, both GN and CN are bolded, even though the gap between GN and CN is quite large. I understand that this might be due to the fact that the difference of the mean results is within the sum of standard deviations (making it unclear whether improvement is statistically significant), but then the same strategy should be applied to other tables/results. For example, going with the same logic, BN should be bolded in Table 11, for Acc in Tiny ImageNet. I would like to ask the authors to state the strategy for deciding when a given method is statistically "best" (i.e. bolded in the paper), use it consistently throughout all the experiments in the paper, and openly discuss the fact that sometimes two methods are "tied" for the 1st place.
> > >
> > > My other reservation is about the motivation of the proposed method, although I don't think it can be addressed in this review process (not enough time). Namely, although the answers provided by the authors to the questions I raised about the intuition behind results (e.g. why GN negates forgetting in BN) are reasonable, they are not backed up by empirical experiments or clear theoretical reasoning. I still don't think that the proposed method is very well understood, and there is more to analyze here. On the other hand, the experimental evaluation is quite extensive as it is.
> > >
> > > Minor point: thank you for providing the value of $\eta$. However, when I suggested replacing the running mean with a standard mean, I meant the cumulative moving average, I apologize for being imprecise. PyTorch has a flag for that, it can be activated by setting the value of momentum to None in `BatchNorm2d`. This is not a major issue, but I still think an analysis of the $\eta$ hyperparameter could be interesting.

---

> > > > ### Author Response · Authors · 2021-11-22
> > > > **Second Response to Reviewer BLEs**
> > > >
> > > > We thank the Reviewer for the prompt reply with the clarification questions.
> > > >
> > > > **Concern #1: presentation of the results**
> > > >
> > > > We apologize for the inconsistency in the presentation of our results. Our work uses the strategy of bolding the best averaged scores, as indicated in the caption of each table. We have amended the presentation of our Tables to follow this rule. We also added Section 6.4 in the main paper to discuss more about the limitations from the experiments, including:
> > > >
> > > > (i) DER++ with GN achieved lower FM compared to CN in Table 2 (R 9JHA);
> > > >
> > > > (ii) high standard deviations in some scenarios such as Table 4 and 11 (R BLEs); and
> > > >
> > > > (iii) CN had lower evaluation metrics on the minority classes, NUS-WIDEseq benchmark (R BLEs).
> > > >
> > > > **Concern #2: the intuition behind the results are not backed up empirically or theoretically.**
> > > >
> > > > Our experiments use three standard evaluation metrics to measure the overall performance (ACC), forgetting (FM), and forward transfer (LA). We believe that the current empirical results support our motivation and discussions thus far: BN facilitates forward transfer (higher LA) compared to GN; GN suffers less forgetting (lower FM) than BN; CN could balance between BN and GN with comparable LA to BN and lower FM than BN. For a more detailed analysis, we refer the Reviewer to our response to Concern #3 of R 8iRV.
> > > >
> > > > **Minor point 1: using the cumulative moving average (CMA)**
> > > >
> > > > Thank you for the clarification. Using the simple CMA would yield undesirable behaviors and results. We first recall that the running moments in BN depend on two factors: (i) the sequence of mini-batches; and (ii) the network parameter at each time of calculating the moments. Therefore, we can write the CMA mean of BN for the first $n$-th mini-batches with all of the dependencies as:
> > > >
> > > > $\mu_{\text{CMA}} = \frac{1}{n} [\mu(b_1, \omega_1) + \ldots + \mu(b_n, \omega_n)] = \frac{1}{n}\sum_{i=1}^n \mu(b_i,\omega_i)$
> > > >
> > > > Where $\mu(b_i, \omega_i)$ denotes the mean of $b_i$ - the $i$-th mini-batch calculated using $\omega_i$ - the network parameter at time $i$. With the CMA approach, the network parameters at the early stages of training contribute equally to CMA moments compared to the latest parameter, which is problematic because the early parameters are not well-trained and do not observe the newer tasks' samples. When using the exponential running average (the standard practice), the contributions of early parameters to the running moments degrades exponentially compared to the recent, well-trained parameters.
> > > >
> > > > It is also interesting to note that our BN* strategy is equivalent to not estimating a running averaged but calculating a global mean at the time of evaluation as:
> > > >
> > > > $\mu_{BN^*} = \frac{1}{n}\sum_{i=1}^n \mu(b_i,\omega_n)$
> > > >
> > > >
> > > > **Minor point 2: analysis of $\eta$**
> > > >
> > > > Thanks for the interesting suggestion. For consistency with the Pytorch framework, we refer to the update of the running statistics in BN as
> > > > $\hat{x}_{\text{new}} = (1 - \eta) \times \hat{x} + \eta \times x_t $,
> > > > where $\hat{x}$ is the current running statistic, and $x_t$ is the current mini-batch statistics. With this formulation, lower $\eta$ values assign higher weights to the running part of the running statistics, while higher $\eta$ values will focus more on recent statistics.
> > > >
> > > > Due to the limited time during the rebuttal process, we conducted a preliminary investigation on the Split CIFAR-100 benchmark with ER under the task-aware setting. The following table reports the results of ER with different BN configurations.
> > > >
> > > > | ER w/ BN |  |  | Split CIFAR-100 |  |
> > > > |:---:|:---:|:---:|:---:|:---:|
> > > > |  |  | ACC | FM | LA |
> > > > | CMA |  | 35.86+/-0.90 | 20.24+/-1.30 | 45.53+/-3.65 |
> > > > | EMA  | $\eta = 0.01$ | 62.90+/-2.03 | **6.32+/-2.32** | 64.81+/-1.15 |
> > > > |  | $\eta = 0.05$ | 63.61+/-1.11 | 7.18+/-1.08 | 68.43+/-0.87 |
> > > > |  | $\eta = 0.1$ (standard) | **64.97+/-1.09** | 9.24+/-1.98 | **71.56+/-0.75** |
> > > > |  | $\eta = 0.5$ | 62.43+/-2.65 | 8.54+/-2.76 | 68.86+/-0.51 |
> > > > |  | $\eta = 0.9$ | 61.46+/-2.36 | 9.03+/-2.38 | 66.81+/-1.49 |
> > > >
> > > > As we discussed, the standard CMA strategy yields undesirable performance and is worse than EMA.
> > > > For BN with EMA, low values of $\eta=0.01 \text{ or } 0.05$ slow the running statistics’ changing rate, which makes the EMA statistics more stable and results in less forgetting (indicated by lower FM). However, these settings cannot take advantage of the statistics from the recent model’s parameters and result in lower forward transfer (lower LA), which results in worse overall performance (lower ACC).
> > > > On the other hand, with  $\eta > 0.1$, the running statistics can be too sensitive to the noisy updates from mini-batches, resulting in worse performance than the standard setting of $\eta = 0.1$.
> > > > We will include a more detailed discussion regarding $\eta$ in the final version. Lastly, any improvements with this parameter can also benefit our CN by changing its BN component accordingly.

---

> > > > > ### Comment · Reviewer_BLEs · 2021-11-22
> > > > > **Response to the Authors**
> > > > >
> > > > > I thank the reviewers for additional experiments, clarifications and extending the discussion of the limitations of the paper. In my opinion the modifications introduced by the authors singificantly improved the quality of the paper. As such, I have decided to increase the score to 8 (accept, good paper).

---

> > > > > > ### Author Response · Authors · 2021-11-26
> > > > > > **Thanks for your feedback**
> > > > > >
> > > > > > We are delighted that our rebuttal has addressed the Reviewer concerns. We also want to thank the Reviewer for the valuable feedback and discussion during the rebuttal period.

---

### Official Review · Reviewer_8iRv · 2021-11-02

**Correctness:** 3
**Technical Novelty And Significance:** 3
**Empirical Novelty And Significance:** 3
**Recommendation:** 6
**Confidence:** 4

**Main Review:**

### Strengths

1- I like how the paper is structured. It is well-motivated and well-written.
2- The arguments are sound, supported by experimental design.


### Weaknesses

1- I expected more details in the earlier sections of the paper regarding the effect of normalization schemes on the role of batchnorm. For instance, in Table 1, it would be interesting to see what would the performance metrics be if we don't use BatchNorm at all. So, generally, I suggest extending Section 4.1 with more emphasis on the performance when we use normalization versus when we don't use normalization.

2- It would have been more interesting to disentangle the role of normalization from replay buffers. Table 1 suggests the benefit of BN* diminishes when we are using the replay method, and Table 2 doesn't report the results on the Naive Finetuning (or Single in the paper). So, if possible, I encourage the authors to add more results (for MNIST and CIFAR) for the Naive(Single) method in the appendix.

3- While the arguments about normalization layers are sound and intuitive, the reasons behind the benefit/drawbacks of different normalization layers are still not studied beyond the fact that the statistics change. I believe it is important to study "how" these changes in statistics change performance. For instance, how would normalization schemes impact the forward/backward transfer?


-----------
### Update After Discussion Period

I believe the authors have addressed my previous comments and I increase my initial score.

**Summary Of The Paper:**

The paper studies the role of normalization layers in continual learning. The main argument is that vanilla BatchNorm is not very suitable since the statistics of data change across tasks. To alleviate this, the authors propose the CntinualNorm layer which is essentially the combination of GroupNorm followed by BatchNorm. The experiments compare different normalization schemes across various CL benchmarks.

**Summary Of The Review:**

I believe the paper is good and intuitive, however needs some improvements that can improve its contributions.

---

> ### Author Response · Authors · 2021-11-18
> **Response to Reviewer 8iRv**
>
> **Concern #1: performance without normalization**
>
> Thanks for the suggestion. We have added the discussion regarding the benefits of BN in Section 4.1 and provided the corresponding results of the no normalization layer (noNL) on the more challenging benchmarks of Split CIFAR-100 and Split mini IMN in Table 2. We observe that with both ER and DER++, the noNL has the lowest evaluation metrics (FM, LA, and ACC) compared to other normalization layers. While low FM suggests that the model does not forget much throughout training, low LA indicates that it struggles to learn new knowledge on these challenging benchmarks, results in poor final final performance in ACC (see our discussion to Concern #3 for more details).
> This result sheds light on how BN and other normalization layers affect continual learning. Although normalization layers facilitate continual learning by improving the learning of individual tasks, they also make the model more prone to catastrophic forgetting since there is more accumulated knowledge to forget.
>
> **Concern #2: decouple CN from replay**
>
> We have included the performance of the naive finetuning (Single) method in Appendix E.6, Table 10. On both Split CIFAR-100 and Split mini IMN benchmarks, we observe consistent improvements of CN over BN in the final performance (1% on Split CIFAR-100 and 2% on Split mini IMN). This improvement comes from the reduction of catastrophic forgetting (around 0.7% reduction in FM) and the improvements of forward transfer (from 1 to 2% increases in LA).
> Please note that this naive strategy is extremely challenging to train in continual learning because it does not have any mechanism to prevent catastrophic forgetting or support forward transfer. We also observe that GN achieved competitive performance on the Split CIFAR-100 benchmark thanks to its ability to alleviate the cross-task normalization effects compared to other methods that have a BN component. However, GN's performance falls short on the more challenging Split mini IMN benchmark because it cannot learn individual tasks well. The increased difficulties in the Split mini IMN benchmark could come from larger image dimensions and more diverse class labels (the classes in the CIFAR-100 benchmark come from 10 super-classes, while the mini IMN classes do not).
>
>
> **Concern #3: How these changes in statistics change the performance. How would normalization schemes impact the forward/backward transfer.**
>
> **Concern #3a: how the statistics change affect the performance**
>
> Thanks for the suggestions. In this work, we measure three aspects of continual learning methods via the following common evaluation metrics: ACC (higher is better) measures the overall performance of all tasks at the end of training; FM (lower is better) measures the averaged performance degrade of older tasks at the end of training, which reflects the degree of catastrophic forgetting; and LA (higher is better) which averages the performance of each task right after the model learned it, reflecting the model’s ability of forward transfer.
>
> **Concern #3b:  How would normalization schemes impact the forward/backward transfer.**
>
> **Forward transfer**
>
> BN is helpful for forward transfer in two aspects. First, BN makes the optimization landscape smoother [1], which allows the optimization of deep neural networks to converge faster and better [2]. In continual learning, BN enables the model to learn individual tasks better than the no-normalization method.
> Second, with the episodic memory, BN uses data of the current task and previous tasks (in the memory) to update its running moments. Therefore, BN further facilitates forward knowledge transfer during experience replay: current task data is normalized using moments calculated from both current and previous samples.
> Compared to BN, spatial normalization layers (such as GN) lack the ability to facilitate forward transfer via normalizing using moments calculated from data of both old and current tasks.
>
> **Backward transfer**
>
> In this work, we discovered that the discrepancy between training and testing modes in BN is one source of causing higher forgetting. Particularly, BN’s running moments could be highly biased compared to the true/global ones because of the continual learning setting.
> Compared to BN, GN performs the same computation during training and testing and is less affected in the presence of data imbalance.
> Therefore, BN can cause higher forgetting than spatial normalization layers such as GN.
>
> In summary, we argue that BN facilitates the model’s ability to transfer knowledge across tasks compared to spatial and existing adaptive normalization layers, which should result in higher LA. At the same time, BN also causes higher forgetting, which should result in higher FM.
> From our experiments across different settings, the results are consistent with this analysis: BN generally has higher FM than both GN and SN while having similar LA to SN and higher LA to GN.

---

> > ### Author Response · Authors · 2021-11-18
> > **Reference**
> >
> > [1] Santurkar, Shibani, et al. "How does batch normalization help optimization?." Proceedings of the 32nd international conference on neural information processing systems. 2018.
> >
> > [2] Bjorck, Johan, et al. "Understanding batch normalization." Proceedings of the 32nd International Conference on Neural Information Processing Systems. 2018.

---

> > > ### Comment · Reviewer_8iRv · 2021-11-20
> > > **Response to authors**
> > >
> > > Thank you for your response. I believe the additional experiment and results have improved the paper and for that, I will increase my initial score.

---

> > > > ### Author Response · Authors · 2021-11-22
> > > > **Response to Reviewer 8iRV**
> > > >
> > > > We are delighted that our response addressed your concerns and we appreciate that the Reviewer has changed the score accordingly.

---

### Author Response · Authors · 2021-11-18
**General Response to all Reviewers**

We thank the Reviewers for insightful comments and valuable feedback. We are delighted that they found our work to be well-motivated (R 8iRV, R 9JHA), the problem studied is highly practical, relevant, and novel under the continual learning setting (R BLEs, R 9jXz, R 9JHA). They (R 8iRV, R BLEs, R 9jXz, R 9JHA) agree that our paper is well-written, the experiments are extensive with mostly consistent improvements. We appreciate the Reviewers’ interesting suggestions and will incorporate all feedback into the final version of our work. In the following, we summarize key changes in the revision in response to the Reviewers’ concerns.

- In the main paper, we amended Section 4.1 to discuss the benefit of BN (text in blue) and provided the results of the no-normalization layer in Table 2.

- We conducted the experiments as suggested by all Reviewers and included the results in Appendix E.4 to Appendix E.8.

- We thank R BLEs and R 9jXz for spotting the mistake in Section 2 and 3. We decided to keep the Section number as it is to keep the reference consistent during the reviewing and rebuttal period. We will merge the current Section 2 and 3 in the final version of this work.

In the following, we will address the concern from each Reviewer individually.

---

### Decision · Program_Chairs · 2022-01-20

**Decision:**

Accept (Poster)

**Comment:**

The paper sheds light on issues with BN in continual learning and proposes a quite simple, which is a strength, solution to fix it.

The Authors first draw attention to the fact that using recalculated moments boosts performance and reduces forgetting, which serves as an argument that at least partially BN contributes to catastrophic forgetting in continual learning. Given that BN remains quite important in certain application areas such as vision, it is a strong motivation for the paper.

The experiments are thorough and clearly show that CN is a practically relevant alternative to BN in continual learning.

One weakness of the paper is that the method is poorly motivated, and relatedly, it has quite limited novelty. CN combines the strengths and weaknesses of BN and GN. Hence, it is not clear why it outperforms both, given that it still has the issue of BN that normalization statistics might become outdated. This is one of the weaknesses pointed out by 9jXz who recommended rejecting the paper. It would be also nice to compare to Mode Normalization https://openreview.net/forum?id=HyN-M2Rctm.

Other papers have suggested changing normalization for sequential learning. Changing batch normalization (to batch renormalization) was investigated in [1] in the context of continual learning. Relatedly, [2] proposes TaskNorm for meta-learning.

Despite these issues, it is a solid contribution and it is my pleasure to recommend acceptance. In the camera-ready, please describe more clearly the design principles behind CN.

[1] Rehearsal-Free Continual Learning over Small Non-I.I.D. Batches, https://openaccess.thecvf.com/content_CVPRW_2020/papers/w15/Lomonaco_Rehearsal-Free_Continual_Learning_Over_Small_Non-I.I.D._Batches_CVPRW_2020_paper.pdf

[2] TaskNorm: Rethinking Batch Normalization for Meta-Learning, https://arxiv.org/abs/2003.03284